# PostAlign: Multimodal Grounding as a Corrective Lens for MLLMs

**Yixuan Wu**[1,2]    **Yang Zhang**[1,3]    **Jian Wu**[2]    **Philip Torr**[1]    **Jindong Gu**[1] [*]
[1]University of Oxford      [2]Zhejiang University      [3]National University of Singapore
`wyx_chloe@zju.edu.cn`

## Abstract

Multimodal Large Language Models (MLLMs) have shown remarkable performance in vision-language tasks, such as image captioning and visual question answering. However, these models often struggle with fine-grained visual understanding and are prone to hallucinations, primarily due to over-reliance on linguistic priors that distract them from leveraging actual visual information. This results in outputs that are often unanchored in the visual content, leading to errors. To address these challenges, we introduce MMGrounded-PostAlign, a post-multimodal alignment framework designed to enhance the visual understanding capabilities of MLLMs and mitigate hallucinations. In the framework, the visual grounding module identifies the referred objects in the image, while the textual grounding module generates the rationale for the final answer. This dual grounding approach ensures that outputs are firmly anchored in both visual and textual evidence. In particular, we incorporate a negative rejection mechanism within the visual grounding module to distinguish between grounded entities and non-existent objects influenced by linguistic biases. Moreover, we propose a selective reasoning mechanism within the textual grounding module to adjust the model's reasoning strategy based on the complexity of the query. These innovations together work to resolve the issues associated with hallucinations and enhance the overall alignment between visual and textual modalities. Extensive evaluations on benchmarks such as POPE, Halo-Quest, ReasonSeg, MME, and MMBench demonstrate significant improvements in fine-grained visual understanding and hallucination suppression, showcasing the effectiveness of our approach in real-world multimodal tasks.

## 1 Introduction

Recently, the rapid development of Multimodal Large Language Models (MLLMs) has significantly advanced visual understanding by improving multimodal alignment between visual and textual representations. This multimodal alignment enables MLLMs to bridge the gap between modalities, leveraging large-scale vision encoders and pretrained language models to achieve remarkable results in tasks such as image captioning, visual question answering, and visually grounded dialogue (Guo et al., 2025; Team, 2024; Wang et al., 2024c; Achiam et al., 2023; Liu et al., 2024b; 2023c; Team et al., 2023; Zhou et al., 2025; Feng et al., 2023; 2024). However, as tasks grow more demanding in terms of fine-grained visual understanding and complex reasoning, these models often fail to maintain robust alignment between modalities (Liu et al., 2024a; Bai et al., 2024; Zhou et al., 2023; Wu et al., 2024), revealing critical limitations that hinder their robustness and reliability.

A key factor behind these limitations is the model's reliance on spurious correlations (Calude & Longo, 2017; Ye et al., 2024) instead of genuine causal or contextual relationships. Rather than interpreting visual content based on meaningful interactions, MLLMs often default to statistical associations, such as objects frequently appearing together in images. This problem is further amplified by linguistic priors (Leng et al., 2024), which shape the model's predictions based on common textual patterns rather than actual visual evidence.

One major issue caused by this bias is hallucination (Liu et al., 2024a; Bai et al., 2024), where the model generates content that does not exist in the image. Instead of relying on visual details, it

---
[*]Corresponding author.

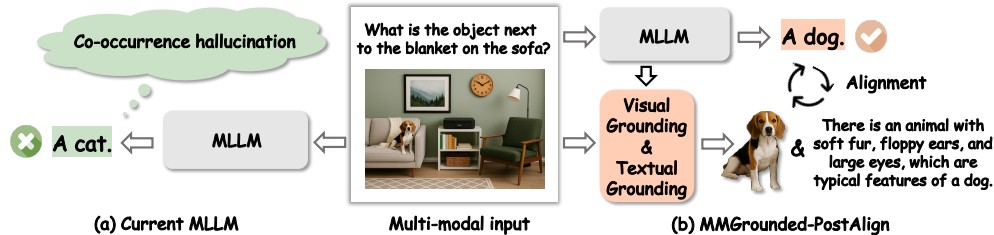

Figure 1: (a) Current MLLMs often struggle with spurious correlations, leading to co-occurrence hallucinations, such as incorrectly identifying a cat on the sofa. (b) To address this, we propose MMGrounded-PostAlign, which incorporates a multimodal grounding module that enhances answer accuracy by aligning and anchoring the final output with actual visual and textual evidence.

selects words that are likely to appear together in text, even if they are incorrect in the given visual context. This reveals a lack of robust multimodal grounding. Another challenge is the over-reliance on high-level visual cues. Instead of recognizing fine-grained object attributes, MLLMs often focus on general patterns like backgrounds or dominant colors. Linguistic priors encourage broad, text-driven generalizations that do not always align with the actual scene. As a result, the model struggles with precise visual understanding. MLLMs also face difficulties in complex reasoning tasks. When interpreting object interactions, answering compositional questions, or inferring causal relationships, they often lack logical consistency. Rather than constructing well-grounded multimodal responses, the model may rely on familiar text patterns, leading to reasoning failures that prioritize language biases over actual visual input.

To address these challenges, MLLMs need stronger multimodal alignment to ensure that textual outputs remain faithful to visual input. Specifically, we propose multimodal grounding as the corrective lens to mitigate these limitations and reduce reliance on spurious correlations. Multimodal grounding improves the alignment between modalities, helping the model establish more reliable connections between language and relevant sources of information. Rather than relying on high-level correlations or co-occurrence-based embeddings, grounding can take various forms. It may link textual descriptions to visual regions, external knowledge bases, or contextual cues. By reinforcing consistency across modalities, grounding helps mitigate hallucinations, reducing the impact of spurious correlations and improving output reliability.

In this paper, we propose MMGrounded-PostAlign, a post-multimodal alignment framework designed to enhance the fine-grained visual understanding capabilities of MLLMs and reduce hallucinations caused by over-reliance on linguistic priors. Our framework incorporates a multimodal grounding module that provides two complementary types of evidence: visual grounding, which identifies the referred object in the image, and textual grounding, which generates an interpretable rationale before producing the final prediction. This dual-grounding design ensures that the model's outputs are firmly anchored in real visual cues and contextually appropriate textual reasoning. Within this framework, we further introduce a negative rejection mechanism in the visual grounding module. This mechanism explicitly distinguishes grounded visual entities from nonexistent objects and enables the model to reject visual regions that are not supported by the image, thereby complementing multimodal grounding in suppressing hallucinations influenced by language priors. On the textual grounding side, we propose a selective reasoning mechanism that dynamically controls whether the model should engage in textual grounding. The key insight is that not all queries require explicit reasoning: for simple cases, the model can directly produce the final answer without constructing an intermediate rationale, whereas for complex cases, the model benefits from performing textual grounding to organize its reasoning steps. To realize this behavior, we introduce two routing signals, <SIMPLE> and <COMPLEX>, which guide the model to selectively invoke textual grounding only when necessary. By integrating these components, our framework reinforces multimodal alignment through both visual and textual evidence, leading to more accurate visual understanding and substantially reduced hallucinations in MLLMs.

Our contributions are summarized as follows:

- We propose MMGrounded-PostAlign, a post-multimodal alignment framework that enhances visual understanding in MLLMs by aligning visual and textual modalities through multimodal grounding, ensuring outputs are anchored to actual visual and textual evidence.

- We introduce a negative rejection mechanism in visual grounding to mitigate hallucinations, allowing the model to differentiate between grounded objects and hallucinated artifacts.

- In textual grounding, we propose a selective reasoning mechanism that adapts the model's reasoning strategy based on the complexity of the input query, improving the model's ability to handle reasoning tasks with various complex levels.

- We evaluate our method on benchmarks such as HaloQuest, POPE, ReasonSeg, MMBench, MME, and RefCOCO, achieving significant improvements in visual understanding and hallucination suppression, while preserving the general reasoning capabilities of MLLMs.

## 2 RELATED WORK

### 2.1 VISION-LANGUAGE GROUNDING MODELS

Grounding in vision-language models involves establishing correspondences between textual descriptions and image regions (Xie et al., 2023). Early grounding methods (Yu et al., 2018; Yang et al., 2019) primarily adopted object detection pipelines, while later models (Kamath et al., 2021; Yao et al., 2022; Li et al., 2022; Liu et al., 2024c) integrated vision-language pretraining for better end-to-end performance. Beyond bounding box grounding, segmentation-based grounding (Wu et al., 2020; 2023; 2022) addressed coarse granularity by enabling pixel-wise segmentation for finer localization.

With the rise of MLLMs, grounding tasks have been integrated into generalist frameworks. Some models, like Kosmos-2 (Peng et al., 2023) and Shikra (Chen et al., 2023b), formulate grounding as text-based bounding box prediction, while others (Zhang et al., 2024a; Zhao et al., 2023a; Lai et al., 2024; Xia et al., 2024; Rasheed et al., 2024; Ren et al., 2024; Zhang et al., 2024b), integrate separate grounding modules into MLLMs. These methods primarily focus on enhancing grounding accuracy within a single-task framework, typically using MLLMs to control or guide grounding. In contrast, our work does not aim to introduce a new grounding architecture. Instead, we repurpose visual grounding, together with an additional textual grounding component, as multimodal evidence for post-alignment. Rather than using MLLMs to perform grounding, we leverage grounding outputs to enhance MLLMs' visual understanding and mitigate hallucinations.

### 2.2 HALLUCINATION IN MLLMS

Hallucination in MLLMs, characterized by contradictions between image input and textual output, has been a prevalent issue (Liu et al., 2024a; Bai et al., 2024). To alleviate hallucinated content, existing works can be divided into the following two directions.

The first focuses on post-processing approaches, including post-hoc corrections (Zhou et al., 2023; Yin et al., 2024; Lee et al., 2023; Zhou et al., 2023) and specialized decoding (Huang et al., 2024; Chen et al., 2024; Leng et al., 2024; Zhu et al., 2024; Zhao et al., 2024; Deng et al., 2024; Wang et al., 2024a). For example, Lee et al. (2023) introduce a self-revising mechanism to reduce hallucination. However, these methods often require increased inference time, limiting their generalizability and scalability across diverse data domains and model sizes.

The second line of work focuses on training-based methods. Some of them focus on data-level improvements, including the introduction of negative data (Liu et al., 2023b), counterfactual data (Yu et al., 2024), and reducing noise in existing datasets (Wang et al., 2024b; Yue et al., 2024). Reinforcement learning (RL) has also been explored to guide MLLMs toward hallucination-free outputs (Zhao et al., 2023b; Li et al., 2023a; Gunjal et al., 2024; Sun et al., 2023). Instead of solely relying on data augmentation or reinforcement learning, our method integrates a multimodal grounding module for stronger multimodal alignment, ensuring that model outputs are firmly anchored in real evidence.

## 3 METHOD

In this section, we present MMGrounded-PostAlign, a post-multimodal alignment framework that integrates both visual grounding and textual grounding as multimodal evidence for correcting hallucinations in MLLMs. Unlike prior grounding-oriented works, our goal is not to introduce a new

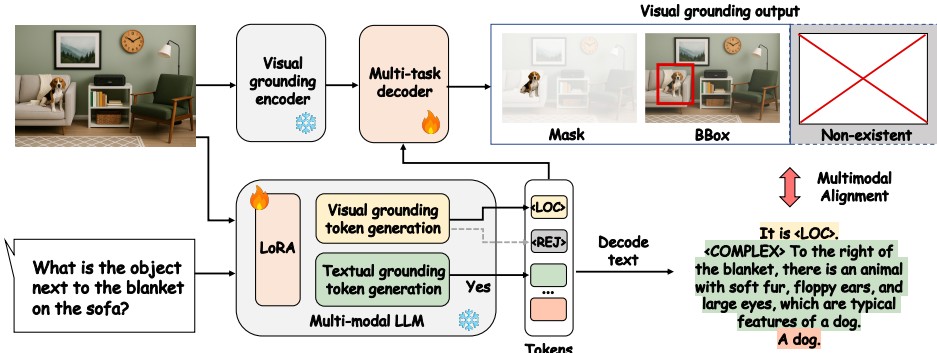

Figure 2: The pipeline of our proposed MMGrounded-PostAlign. Given an image and text query, the MLLM generates the tokens of visual grounding (<LOC>), textual grounding, and the final answer. The last-layer embedding of <LOC> is fed to multi-task decoder and then decoded into a segmentation mask and bounding box for the target object. When no target object exists in the image, the visual grounding token is replaced by <REJ>, which is directly assigned an empty mask and bounding box in the multi-task decoder. The textual grounding facilitates the generation of a rationale prior to the final answer for complex queries.

grounding architecture. Instead, we repurpose grounding outputs as a corrective lens: visual grounding provides image-grounded cues to reduce the MLLM's reliance on linguistic priors, while textual grounding anchors the reasoning process with interpretable rationales. Together, these two signals enhance fine-grained visual understanding and guide the MLLM toward evidence-aligned predictions.

## 3.1 FRAMEWORK OVERVIEW

As shown in Figure 2, our framework comprises three key components: (1) A MLLM that processes image and text inputs to predict the tokens of visual grounding, textual grounding, and the final answer. (2) A visual grounding encoder that extracts image features. (3) A multi-task decoder that performs visual grounding of segmentation mask and detection bounding box, using the visual grounding token as prompts.

**Multi-modal Input and Structured Output.** Given an image $\mathcal{I}$ and a text query $\mathcal{Q}$, the MLLM is tasked with generating a structured output $\mathcal{A}$, as:

$$\mathcal{A} = \text{MLLM}(\mathcal{I}, \mathcal{Q}), \tag{1}$$

$$\mathcal{A} = \{\mathcal{V}, \mathcal{T}, \mathcal{F}\}, \tag{2}$$

where $\mathcal{V}$ denotes the visual grounding token, $\mathcal{T}$ represents the textual grounding token for rationale, and $\mathcal{F}$ is the final answer token.

**Visual Grounding Prediction.** The last-layer embedding of the visual grounding token <LOC> is extracted from the MLLM and passed through an MLP projection layer to obtain the prompt embeddings for the subsequent visual grounding. Meanwhile, the visual grounding encoder extracts dense visual features from the input image. These dense visual features, along with the prompt embeddings, are fed into the multi-task decoder, which generates the final visual grounding outputs.

**Grounding-Augmented Final Answer Generation.** The visual grounding outputs and textual grounding rationales act as implicit constraints that guide the MLLM's final answer generation. The textual grounding provides a structured framework for generating interpretable rationales, which, when integrated with the visual grounding cues, anchor the model's reasoning in both visual and linguistic contexts. This guidance helps mitigate the model's over-reliance on linguistic priors, enhancing its ability to perceive and understand visual information more accurately.

## 3.2 REJECTING NON-EXISTENT OBJECTS

Despite the advanced understanding capabilities of MLLMs, they are often prone to hallucinating non-existent objects due to strong linguistic priors and co-occurrence biases present in the training

data. These hallucinations manifest in various forms, including incorrect attributes, inaccurate spatial locations, or entirely fabricated objects.

To mitigate this, we introduce a negative rejection mechanism, which enables the model to explicitly reject non-existent objects rather than hallucinating their presence. Specifically, if the referents are absent from the image, we enforce the MLLM to predict a special `<REJ>` token to replace the corresponding `<LOC>` token. In the multi-task decoder, the predicted `<REJ>` tokens are directly assigned empty masks and bounding boxes, effectively bypassing the decoding process. The primary advantage of the negative rejection mechanism is its ability to mitigate the model's over-reliance on linguistic priors. Many MLLMs tend to associate high-frequency co-occurring objects (e.g., predicting a "*table*" when a "*chair*" is present) and produce incorrect outputs. By explicitly learning to reject such negative samples, the model becomes more capable of distinguishing between genuine visual grounding and misleading language priors. Additionally, incorrect attributes and spatial misplacement, such as predicting a "*red apple*" when only green apples are present, are also suppressed, as the model is penalized for assigning masks or bounding boxes to absent entities.

**Negative Rejection Loss.** To further reinforce the model's ability to reject hallucinated objects, we introduce a negative rejection loss, which explicitly penalizes incorrect grounding predictions when the referent is absent. The loss is defined as:

$$\mathcal{L}_{\text{rej}} = -\frac{1}{N} \sum_{i=1}^{N} \left[ y_i^{\text{rej}} \log p_i^{\text{rej}} + (1 - y_i^{\text{rej}}) \log(1 - p_i^{\text{rej}}) \right], \tag{3}$$

where $y_i^{\text{rej}} \in \{0, 1\}$ denotes whether the referent in sample $i$ should be rejected.

### 3.3 Selective Reasoning in Textual Grounding

Previous work (Zhang et al., 2023) has shown that generating intermediate rationales through textual grounding improves the reasoning capabilities of MLLMs, leading to improved answer accuracy for complex tasks. In our multimodal grounding-assisted MLLM framework, we argue that not all queries require explicit rationale generation via textual grounding. Instead, a selective approach is needed to balance the visual output from the visual grounding module with the textual answer generation of the MLLM. Our experiments, as shown in Table 3, also support this hypothesis. To this end, we propose a selective reasoning mechanism, which allows the MLLM to determine whether explicit contextual reasoning is needed. Specifically, during training, we categorize queries into `<SIMPLE>` and `<COMPLEX>` types:

- For simple queries, the MLLM directly outputs the visual grounding token and the final answer, skipping rationale generation. For example, for the simple query: *"User: <IMAGE> What color is the car in this image?" "Assistant: It is <LOC>. <SIMPLE>. The car is red."*

- For complex queries, the MLLM's output includes the visual grounding token, the contextual rationale, and the final answer. For example, for the complex query: *"USER: <IMAGE> Which food in the picture contains the most protein?" "ASSISTANT: It is <LOC>. <COMPLEX>. This image contains oranges, eggs, vegetables, and buns. Among them, eggs are the richest in protein. "*

To enable automatic complexity assessment during inference, we incorporate self-reflection prompting into the user instructions:

> *Given an image and a text query, first assess whether answering the query requires a rationale. If the answer is directly observable from the image without additional reasoning, classify it as <SIMPLE> and provide only the final answer. If answering requires logical inference or contextual understanding, classify it as <COMPLEX> and generate a rationale before providing the final answer.*

**Selective Reasoning Loss.** We also design a selective reasoning loss to supervise the model in identifying whether a query requires complex reasoning. Each query is classified as either `<SIMPLE>`

or `<COMPLEX>`, and the loss is computed as:

$$\mathcal{L}_{\text{reason}} = -\frac{1}{N} \sum_{i=1}^{N} \left[ y_i^{\text{rea}} \log p_i^{\text{rea}} + (1 - y_i^{\text{rea}}) \log(1 - p_i^{\text{rea}}) \right], \quad (4)$$

where $y_i^{\text{rea}} \in \{0, 1\}$ indicates whether query $i$ requires generating a rationale.

### 3.4 TRAINING OBJECTIVE

In summary, we finetune the MLLMs using LoRA Hu et al. (2022) while jointly optimizing the multi-task decoder. The overall loss function is defined as:

$$\mathcal{L} = \lambda_1 \mathcal{L}_{\text{rej}} + \lambda_2 \mathcal{L}_{\text{reason}} + \mathcal{L}_{\text{ground}} + \mathcal{L}_{\text{text}}, \quad (5)$$

where the visual grounding loss $\mathcal{L}_{\text{ground}}$ consists of a joint optimization of detection loss $\mathcal{L}_{\text{det}}$ and segmentation loss $\mathcal{L}_{\text{seg}}$, formulated as:

$$\mathcal{L}_{\text{det}} = \mathcal{L}_{\text{smooth-L1}}(\hat{y}_{\text{bbox}}, y_{\text{bbox}}) + \mathcal{L}_{\text{GIoU}}(\hat{y}_{\text{bbox}}, y_{\text{bbox}}), \quad (6)$$

$$\mathcal{L}_{\text{seg}} = \mathcal{L}_{\text{BCE}}(\hat{y}_{\text{mask}}, y_{\text{mask}}) + \mathcal{L}_{\text{DICE}}(\hat{y}_{\text{mask}}, y_{\text{mask}}), \quad (7)$$

and $\mathcal{L}_{\text{text}}$ is the cross entropy language modeling loss, defined as:

$$\mathcal{L}_{\text{text}} = \mathcal{L}_{\text{LM}}(\hat{y}_{\text{txt}}, y_{\text{txt}}). \quad (8)$$

## 4 EXPERIMENT

### 4.1 SETUPS

**Training Data Formulation.** We construct a diverse multimodal training dataset, where each instance is annotated with a reasoning-type token: `<SIMPLE>` or `<COMPLEX>`. The `<SIMPLE>` label denotes low-complexity queries requiring minimal reasoning, while `<COMPLEX>` is used for tasks involving higher-order or indirect reasoning. Additionally, negative samples are included with the `<REJ>` token, indicating the absence of a visual referent. Detailed data construction and labeling procedures are provided in the *Appendices*.

**Network Architecture.** Our framework is built upon LLaVA-1.5-7B, LLaVA-1.5-13B (Liu et al., 2024b), Qwen2-VL-7B (Wang et al., 2024c), Qwen2.5-VL-7B (Bai et al., 2025), InternVL3-14B (Zhu et al., 2025), and InternVL3.5-14B (Wang et al., 2025) as the MLLM backbone, and ViT-H SAM (Kirillov et al., 2023) as the visual grounding backbone. To preserve the pre-trained MLLM's knowledge, we employ LoRA (Hu et al., 2022) for parameter-efficient fine-tuning, while freezing the visual grounding encoder. The multi-task decoder, along with the LLM token embeddings, LLM head, and projection layer, is fully fine-tuned. The decoder includes both the mask and bounding box decoders. The hidden embeddings of visual grounding tokens (e.g., `<LOC>`) serve as prompts for SAM, conditioning its mask decoder to predict object masks, while a lightweight MLP regressor predicts bounding box coordinates from SAM-extracted features.

### 4.2 MAIN RESULTS AND ANALYSIS

In this section, we present results across three benchmark categories: (1) hallucination datasets (HaloQuest (Wang et al., 2024d) and POPE (Li et al., 2023b)), (2) generalization and reasoning datasets (MMBench (Liu et al., 2024d), MME (Liang et al., 2024)), and (3) grounding tasks (RefCOCO series (Kazemzadeh et al., 2014) and ReasonSeg (Lai et al., 2024)). Our method demonstrates that, while preserving the general reasoning capabilities of MLLMs, it not only mitigates hallucinations but also enhances visual understanding. We then highlight five key findings that identify the primary sources of hallucinations and demonstrate how our method addresses them.

**Finding 1: Linguistic Priors Override Visual Information in MLLMs.** Figure 3 investigates how MLLMs rely on linguistic priors over visual inputs. We analyze layer-wise token probabilities during decoding in (a)-(b) and the impact of visual input on token generation in (c)-(d) (see *Appendices* for details).

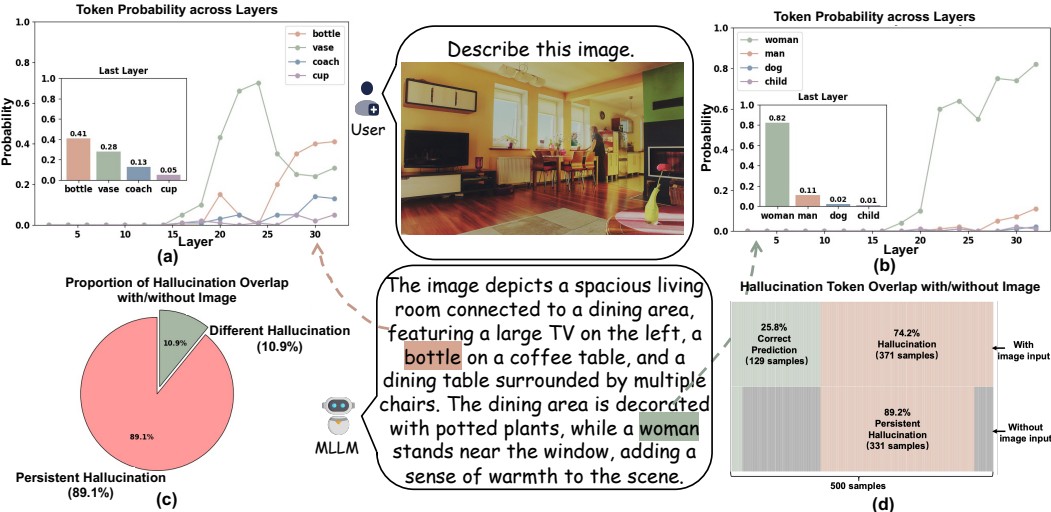

Figure 3: (a)(b) Token probability distributions across transformer layers, showing distinct trends for hallucinated (pink) and non-hallucinated (green) tokens. (c)(d) Removing visual input yields an 89.2% overlap in hallucinated tokens, supporting the hypothesis that hallucinations arise from linguistic biases rather than visual misinterpretation.

In (a) and (b), we examine the Top-4 tokens ranked by probability in the final decoding layer. Non-hallucinated tokens like "woman" have high probabilities from the 20th layer, while hallucinated tokens like "bottle" emerge around the 30th layer. Notably, in (a), the probability of the ground-truth token "vase" drops sharply after the 25th layer, falling below hallucinated tokens. We hypothesize that linguistic priors gradually suppress visual information, steering the model towards language-driven generation.

In (c) and (d), we validate this by removing the image input and analyzing hallucinated token persistence. Across 500 MSCOCO images, the hallucinated token overlap rate is 89.2%. In (d), when the image input is removed, 89.2% of hallucinations persist, confirming that linguistic priors dominate hallucination generation, often overriding visual evidence.

**Finding 2: Explicit Visual Grounding Mitigates Hallucinations.** To investigate the extent to which the visual grounding mechanism enhances MLLM's visual understanding and mitigates visual hallucinations, we first conduct an ablation study on the HaloQuest benchmark. As shown in Table 1, the first row presents the baseline where the visual grounding module is entirely removed. The results indicate that the baseline exhibits poor performance, particularly in the *False Premise* and *Insufficient Context* categories. Introducing the mask grounding (denoted as <SEG>) and the bounding box grounding (denoted as <DET>) significantly improves performance. Furthermore, when both mask grounding and bounding box grounding are incorporated, the MLLM achieves even better results. Notably, the introduction of the <REJ> token for negative rejection further suppresses hallucinations, especially in the *False Premise* and *Insufficient Context* categories. This is primarily because these categories contain a substantial number of anti-concept scenarios. For example, a question such as *"What breed of dog is in the image?"* may have a ground truth response of *"There is no dog in the image."*, which directly evaluates the MLLM's ability to reject misleading premises.

Additionally, as shown in Table 2, we further investigate different grounding strategies. A common approach to integrating MLLM with grounding mechanisms is treating **B**ounding boxes as language **T**okens for unified **L**earning (BTL). Based on this strategy, we explore two training paradigms: **(1) BTL-Generation:** The input consists of an image and a referring text, while the MLLM directly generates the bounding box coordinates. **(2) BTL-Caption:** The input is an image, and the output is a caption describing the image content, where the target object's bounding box is embedded within the generated text. Here, **baseline** refers to our proposed framework with the visual grounding module removed, while retaining other designs (i.e., selective reasoning). The results on the POPE hallucination benchmark indicate that incorporating BTL-Generation does not yield performance gains, while BTL-Caption provides moderate improvement. Combining both approaches further enhances performance. However, our proposed explicit visual grounding module significantly

Table 1: An ablation study evaluates the impact of visual grounding tokens (`<LOC>`) on the HaloQuest benchmark Wang et al. (2024d), where LLaVA-1.5-7B serves as the MLLM baseline. Here, `<SEG>` and `<DET>` are specialized types of `<LOC>` tokens, where `<SEG>` generates segmentation masks and `<DET>` produces bounding boxes. The results show that combining `<SEG>` and `<DET>` enhances visual understanding, while `<REJ>` effectively suppresses hallucinations in misleading or insufficient context scenarios.

| Method | | | False Premise | | Visually Challenging | | Insufficient Context | |
|---|---|---|---|---|---|---|---|---|
| `<SEG>` | `<DET>` | `<REJ>` | Human Eval | Auto-Eval | Human Eval | Auto-Eval | Human Eval | Auto-Eval |
| ✗ | ✗ | ✗ | 2.0 | 2.3 | 23.5 | 23.0 | 2.5 | 1.7 |
| ✓ | ✗ | ✗ | 6.5 | 7.2 | 30.1 | 30.6 | 7.4 | 8.2 |
| ✗ | ✓ | ✗ | 8.2 | 8.9 | 31.1 | 31.7 | 6.6 | 7.4 |
| ✓ | ✓ | ✗ | 9.9 | 10.5 | 33.9 | 35.0 | 9.9 | 11.6 |
| ✓ | ✓ | ✓ | **33.2** | **33.9** | **38.3** | **37.2** | **31.4** | **32.2** |

Table 2: Experiments on the POPE, MME, and MMBench. Results show that explicit visual grounding module outperforms BTL-based methods by enforcing stronger multimodal alignment.

| | POPE( Li et al.) | | | MME | MMBench( Liu et al.) | |
|---|---|---|---|---|---|---|
| | Ran | Pop | Adv | (Liang et al.) | EN | CN |
| LLaVA-1.5-7B | 83.3 | 80.1 | 78.2 | 1504.6 | 62.2 | 57.7 |
| + BTL-Generation | 82.7 | 80.3 | 79.2 | 1489.4 | 59.2 | 54.2 |
| + BTL-Caption | 84.5 | 81.0 | 79.9 | 1499.6 | 59.7 | 54.5 |
| LLaVA-1.5-PostAlign-7B | **86.6** | **84.2** | **82.3** | **1514.3** | **63.9** | **58.7** |
| LLaVA-1.5-13B | 85.4 | 82.2 | 79.2 | 1517.4 | 66.8 | 62.2 |
| + BTL-Generation | 84.4 | 80.9 | 78.3 | 1501.7 | 65.9 | 62.4 |
| + BTL-Caption | 85.1 | 81.7 | 78.8 | 1509.2 | 65.2 | 61.4 |
| LLaVA-1.5-PostAlign-13B | **88.9** | **87.3** | **85.6** | 1520.3 | **68.9** | **63.2** |
| Qwen2-VL-7B | 88.9 | 86.8 | 84.6 | 1717.4 | 82.4 | 79.4 |
| Qwen2-VL-PostAlign-7B | **90.3** | **89.2** | **87.1** | **1729.9** | **83.9** | **80.2** |
| Qwen2.5-VL-7B | 87.3 | 85.1 | 83.4 | 1736.8 | 82.1 | 82.3 |
| Qwen2.5-VL-PostAlign-7B | **89.7** | **87.9** | **85.2** | **1750.2** | **83.3** | **82.9** |
| InternVL3-14B | 89.1 | 87.2 | 84.3 | 1762.8 | 84.3 | 83.1 |
| InternVL3-PostAlign-14B | **91.2** | **89.0** | **86.6** | **1772.3** | **85.5** | **84.3** |
| InternVL3.5-14B | 87.1 | 85.4 | 83.8 | 1792.2 | 82.9 | 82.1 |
| InternVL3.5-PostAlign-14B | **90.6** | **88.5** | **86.2** | **1906.9** | **84.1** | **83.4** |

outperforms these methods. By enforcing a higher degree of multimodal alignment and effectively rejecting negative samples, our method substantially reduces object and attribute hallucinations.

**Finding 3: Our MLLM Retains Strong Reasoning and Generalization Abilities.** In Table 2, we further investigate whether explicitly incorporating visual outputs affects the general reasoning capabilities of MLLMs. To evaluate this, we test our method on several widely-used datasets for generalization and reasoning, including MME, MMBench, and POPE. Our results reveal that our post-alignment method does not degrade the model's reasoning ability compared to the baseline MLLM. In contrast, we observe that the BTL-Generation training paradigm results in a notable decline in the baseline MLLM's reasoning ability. This suggests that the approach used in BTL-Generation may inadvertently bias the model towards overfitting to visual bounding box information at the cost of abstract reasoning, leading to a drop in generalization performance.

**Finding 4: Selective Reasoning in Textual Grounding Optimizes Efficiency and Accuracy.** Previous work (Zhang et al., 2023) has shown that generating intermediate rationales through textual grounding improves the reasoning capabilities of MLLMs, leading to improved answer accuracy for complex tasks. In Table 3, we compare three textual grounding strategies: **(1) Pre-Reasoning:** A separate MLLM (the same as main MLLM) generates a rationale, which is then fed to the main MLLM for final answer generation. **(2) Inter-Reasoning:** The main MLLM integrates reasoning, simultaneously generating rationale and final answers for each query. **(3) Selective-Reasoning:** Dynamically assesses query complexity and generates rationale only for complex queries. Here, **baseline** refers to our proposed framework with the selective reasoning strategy removed, while retaining the visual grounding module. Experimental results indicate that pre-reasoning strategy achieves better performance than inter-reasoning strategy; However, it involves multiple inference

Table 3: An ablation study on ReasonSeg Lai et al. (2024) shows that selective reasoning in textual grounding performs best by adapting to different query complexity, which avoids overthinking on simple queries and ensures sufficient reasoning for complex ones.

| Method | Easy | | Medium | | Hard | |
|---|---|---|---|---|---|---|
| | gIoU | cIoU | gIoU | cIoU | gIoU | cIoU |
| LLaVA-1.5-7B + SAM-ViT-H | 67.7 | 66.4 | 51.2 | 50.2 | 47.0 | 46.3 |
| + pre-reasoning (PR) | 67.3 | 66.7 | 57.2 | **58.1** | 57.0 | **58.3** |
| + inter-reasoning (IR) | 64.3 | 64.7 | 55.5 | 56.3 | 53.9 | 54.8 |
| + selective reasoning (SR) | **68.9** | **67.2** | **58.9** | 57.2 | **57.2** | 57.7 |
| LLaVA-1.5-13B + SAM-ViT-H | 69.2 | 70.3 | 55.2 | 56.2 | 51.7 | 52.2 |
| + pre-reasoning (PR) | 69.7 | 69.3 | 62.7 | 62.2 | 61.9 | 61.2 |
| + inter-reasoning (IR) | 67.2 | 68.1 | 60.9 | 59.2 | 58.2 | 57.2 |
| + selective reasoning (SR) | **70.8** | **71.3** | **64.2** | **65.2** | **62.9** | **63.8** |
| LLaVA-1.5-7B + SAM-ViT-B + SR | 67.6 | 66.0 | 57.3 | 55.5 | 55.9 | 56.3 |
| LLaVA-1.5-7B + SAM-ViT-L + SR | 68.5 | 66.9 | 58.2 | 56.8 | 57.0 | 57.1 |
| LLaVA-1.5-7B + SAM-ViT-H + SR | **68.9** | **67.2** | **58.9** | **57.2** | **57.2** | **57.7** |

Table 4: Performance comparison on the REC and RES tasks.

| Models | RefCOCO | | RefCOCO+ | | RefCOCOg | |
|---|---|---|---|---|---|---|
| | REC | RES | REC | RES | REC | RES |
| Kosmos-2 (Peng et al., 2023) | 52.3 | – | 45.4 | – | 60.5 | – |
| LISA-7B (ft) (Lai et al., 2024) | – | 74.9 | – | 65.1 | – | 67.9 |
| LLaVASeg-7B (ft) (Yang et al., 2024) | – | 76.2 | – | 65.7 | – | 69.8 |
| MiniGPT v2-7B (Chen et al., 2023a) | 88.0 | – | 79.5 | – | 84.1 | – |
| Shikra-7B (Chen et al., 2023b) | 87.0 | – | 81.6 | – | 82.2 | – |
| Ferret-7B (You et al., 2023) | 87.4 | – | 80.7 | – | 83.9 | – |
| LLaVA-Grounding-7B (Zhang et al., 2024a) | 89.1 | 77.1 | **81.6** | 68.7 | 84.8 | 71.5 |
| VisionLLM v2 (Wu et al., 2025) | 87.9 | 76.6 | 77.6 | 64.5 | 82.9 | 70.7 |
| PixelLLM (Ren et al., 2024) | – | 73.0 | – | 66.3 | – | 69.3 |
| GLaMM (Rasheed et al., 2024) | – | 79.5 | – | **72.6** | – | 74.2 |
| GSVA-7B (ft) (Xia et al., 2024) | 86.2 | 77.2 | 72.8 | 65.9 | 81.5 | 72.7 |
| LLaVA-1.5-PostAlign-7B | 88.2 | 77.9 | 78.4 | 68.2 | 83.3 | 73.2 |
| LLaVA-1.5-PostAlign-13B | **89.2** | **79.7** | 80.1 | 70.9 | **85.3** | **74.8** |

rounds, which reduces overall efficiency. Our proposed selective-reasoning approach not only maintains strong performance but also operates with a single inference round. Also, we augment the original benchmark queries into three difficulty levels (easy, medium, hard) to study the relationship between reasoning strategies and query complexity (see *Appendices* for details). We observe that for simple queries, excessive reasoning can slightly degrade performance due to overthinking. Conversely, for complex queries, explicit reasoning is necessary to ensure accurate predictions.

**Finding 5: Our Architecture Provides a Potential Approach for Visual Grounding with MLLMs.** We also evaluate our method's visual grounding performance on the RefCOCO series and ReasonSeg benchmarks. The results surprisingly show that our framework not only improves the MLLM's visual understanding through the visual grounding module, but also that the MLLM can assist the visual grounding module in enhancing its zero-shot grounding capabilities. This suggests a potential approach for visual grounding empowered by MLLMs. However, it is important to emphasize that the primary goal of our work is not to optimize for visual grounding tasks directly. Instead, we aim to leverage multimodal grounding (including both visual and text grounding) as evidence to suppress hallucinations in MLLMs and improve their visual understanding. Consequently, we do not increase the amount of grounding-specific training data to further optimize performance on grounding benchmarks. In Table 4, for the REC and RES tasks, our approach achieves competitive results compared to specialized MLLM-based grounding models. This highlights the superior pixel-level grounding capabilities of our method. Additional results on other visual grounding tasks can be found in the *Appendices*.

## 4.3 QUALITATIVE ANALYSIS

To investigate how grounding influences the model's attention to image features, we visualize the average attention weights assigned to image features over 500 MSCOCO samples during response generation under two conditions: without grounding and with grounding. To emphasize the most salient attention patterns and reduce noise, we apply Top-k Mean Pooling at three levels: For each output token, we select the top-3 highest attention values across all hidden layers and compute their mean. We then aggregate the top-3 at-

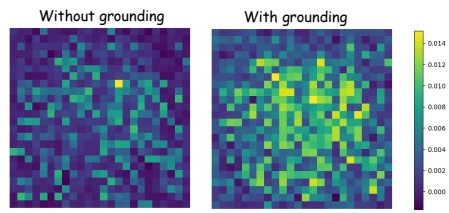

Figure 4: Average attention to image features without (left) and with grounding (right).

tention values across all self-attention heads for each token. Finally, to ensure consistency between the two conditions, we average the top-l attention values across output tokens, where l is the minimum number of tokens generated between the without grounding and with grounding cases. The aggregated attention maps in Figure 4 show a clear distinction between the two conditions. The without grounding attention map exhibits sparse and scattered attention. In contrast, the with grounding map shows significantly more concentrated attention, particularly over relevant image areas. This suggests that grounding effectively guides the model to align textual outputs with pertinent visual features, resulting in more visually grounded responses.

## 5 CONCLUSION

In this paper, we propose MMGrounded-PostAlign, a post-multimodal alignment framework designed to enhance the visual understanding capabilities of MLLMs. Our approach addresses the challenges of over-reliance on spurious correlations by integrating a grounding module that enables both visual and textual grounding. This ensures that the model's outputs are firmly anchored in actual visual and textual evidence. We propose a negative rejection mechanism within the visual grounding module to mitigate hallucinations caused by linguistic biases, helping the model distinguish grounded objects from non-existent ones. Additionally, our selective reasoning in textual grounding adapts the model's reasoning strategy based on query complexity, allowing for more accurate and contextually relevant responses for tasks of varying complexity levels. Extensive evaluations on POPE, HaloQuest, MME, MMBench, ReasonSeg, and RefCOCO demonstrate the effectiveness of our approach in improving visual understanding, reducing hallucinations, and enhancing grounding accuracy.

## ACKNOWLEDGMENTS

This research was partially supported by National Key Research and Development Program of China under Grant No. 2025YFC2423903, National Natural Science Foundation of China under grants No. T2541004, Zhejiang Key R&D Program of China under grant No. 2025C02120, and State Key Laboratory of Transvascular Implantation Devices under Grant No. SKLTID2024003.

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

## A    TRAINING DATA AND LABELS

Our training data composition is summarized in Table 5. The annotation of each query with either `<SIMPLE>` or `<COMPLEX>` is based on the inherent reasoning difficulty and task design of the source dataset. Queries originating from datasets such as COCO-Stuff, ADE20K, Mapillary Vistas, PACO-LVIS, and COCO are typically straightforward and refer to concrete categories without requiring complex reasoning; thus, they are labeled as `<SIMPLE>`. On the other hand, datasets like ReasonSeg, RefCOCO, RefCOCO+, RefCOCOg, RefCLEF, gRefCOCO, Visual Genome, and LLaVA-v1.5-mix665k contain expressions that demand contextual reasoning, ambiguity resolution, or commonsense inference, and are therefore annotated as `<COMPLEX>`. Furthermore, to train the model to explicitly reject invalid or hallucinated referents, we incorporate negative examples marked with the `<REJ>` token. These negative samples are sourced from the gRefCOCO dataset, which includes 32,202 expressions that refer to non-existent targets in the image. This enables the model to learn to distinguish between visually grounded content and spurious linguistic cues. All samples are formatted following the LLaVA-style instruction tuning template to facilitate effective multimodal alignment.

Table 5: Overview of Tasks, Reasoning Token, and Datasets.

| Task | Reasoning Token | Datasets |
|---|---|---|
| Semantic Segmentation | `<SIMPLE>` | COCO-Stuff, ADE20K, Mapillary, LVIS-PACO |
| Referring Segmentation | `<COMPLEX>` | ReasonSeg, RefCOCO, RefCOCO+, RefCOCOg, RefCLEF, gRefCOCO |
| Object Detection | `<SIMPLE>` | COCO |
| Referring Comprehension | `<COMPLEX>` | Visual Genome, RefCOCO, RefCOCO+, RefCOCOg, RefCLEF, gRefCOCO |
| Dialogue | `<COMPLEX>` | LLaVA-v1.5-mix665k |

## B    HALLUCINATED TOKEN ANALYSIS

**Hallucinated Token Extraction.**    To investigate the influence of linguistic priors on hallucinations in MLLMs, we randomly sample 500 images from the MSCOCO dataset Lin et al. (2014). For each image, we generate open-ended responses using the prompt, "*Describe the image.*" in order to elicit detailed descriptions from the model. These responses often contain object tokens that are not present in the image, which we define as hallucinated object tokens. A hallucinated token is any object that is mentioned in the generated text but does not correspond to any visual evidence in the image. The process of identifying hallucinated tokens involves a few key steps. First, we scan the generated response for object names (such as "car", "dog", "table", etc). Next, we cross-reference each identified object token with the objects present in the corresponding image using an object detection model. If an object mentioned in the response is not detected in the image, we categorize it as a hallucinated token.

Moreover, we also extract the textual context immediately preceding each hallucinated token. For example, if the model generates the phrase "*Additionally, there is a car.*" but no car is visible in the image, the hallucinated token would be "car", and its preceding context would be the phrase "*Additionally, there is a*".

**Hallucinated Token Overlap Analysis.**    In order to quantify the influence of linguistic priors on hallucinations, we perform a experiment in which we re-input only the preceding textual context (without any visual input) into the model. The key metric, hallucinated token overlap rate, is calculated as the proportion of hallucinated samples in which the hallucinated tokens from the text-image combination also appear in the set generated from the preceding text alone. This overlap rate gives us an insight into how much influence the linguistic priors have on the hallucination generation process, independent of visual evidence. The hallucinated token overlap rate is computed as:

$$\text{Overlap Rate} = \frac{\text{Number of hallucinated samples without image input}}{\text{Total number of initially hallucinated samples}}. \tag{9}$$

A higher overlap rate indicates that hallucinated tokens persist even when the image is removed, suggesting that these hallucinations are more heavily influenced by the linguistic priors within the model rather than by the visual evidence.

Table 6: Performance comparison on ReasonSeg Lai et al. (2024) benchmark.

| Method | val | | test | | | | | | | |
|--------|-----|-----|------|------|------|------|------|------|------|------|
| | overall | | short query | | long query | | overall | | | |
| | gIoU | cIoU | gIoU | cIoU | gIoU | cIoU | gIoU | cIoU | | |
| OVSeg (Liang et al., 2023) | 28.5 | 18.6 | 18.0 | 15.5 | 28.7 | 22.5 | 26.1 | 20.8 | | |
| GRES (Liu et al., 2023a) | 22.4 | 19.9 | 17.6 | 15.0 | 22.6 | 23.8 | 21.3 | 22.0 | | |
| X-Decoder (Zou et al., 2023a) | 22.6 | 17.9 | 20.4 | 11.6 | 22.2 | 17.5 | 21.7 | 16.3 | | |
| SEEM (Zou et al., 2023b) | 25.5 | 21.2 | 20.1 | 11.5 | 25.6 | 20.8 | 24.3 | 18.7 | | |
| Grounded-SAM (Liu et al., 2024c) | 26.0 | 14.5 | 17.8 | 10.8 | 22.4 | 18.6 | 21.3 | 16.4 | | |
| LISA-7B Lai et al. (2024) | 44.4 | 46.0 | 37.6 | 34.4 | 36.6 | 34.7 | 36.8 | 34.1 | | |
| LISA-7B (ft) Lai et al. (2024) | 52.9 | 54.0 | 40.6 | 40.6 | 49.4 | 51.0 | 47.3 | 48.4 | | |
| LISA-13B Lai et al. (2024) | 48.9 | 46.9 | 39.9 | 43.3 | 46.4 | 46.5 | 44.8 | 45.8 | | |
| LISA-13B (ft) Lai et al. (2024) | 56.2 | 62.9 | 44.3 | 42.0 | 54.0 | 54.3 | 51.7 | 51.1 | | |
| LISA++-7B-LLaVA1.5 (ft) Yang et al. (2023) | 64.2 | 68.1 | 49.6 | 51.1 | 59.3 | 61.7 | 57.0 | 59.5 | | |
| LLaVASeg-7B Yang et al. (2024) | 54.8 | 49.9 | – | – | – | – | – | – | | |
| LLaVASeg-13B Yang et al. (2024) | 59.1 | 52.8 | – | – | – | – | – | – | | |
| GSVA-7B (ft) Xia et al. (2024) | 50.5 | 56.4 | – | – | – | – | – | – | | |
| GROUNDHOG-7B Zhang et al. (2024b) | 56.2 | – | – | – | – | – | – | – | | |
| MMGrounded-PostAlign-7B | 58.3 | 59.9 | 44.5 | 43.9 | 54.1 | 55.7 | 52.0 | 52.9 | | |
| MMGrounded-PostAlign-7B (ft) | 64.4 | 66.2 | 49.2 | 50.3 | 60.1 | 61.2 | 58.2 | 58.9 | | |
| MMGrounded-PostAlign-13B | 63.7 | 65.5 | 50.6 | 48.1 | 59.9 | 62.8 | 57.0 | 58.2 | | |
| MMGrounded-PostAlign-13B (ft) | **69.2** | **73.8** | **55.2** | **54.7** | **66.2** | **64.2** | **63.2** | **62.9** | | |

## C REASONSEG BENCHMARK AUGMENTATION

To investigate the relationship between reasoning strategies and query difficulty, we augment the queries in the ReasonSeg benchmark. For the **easy** category, we simplify the text descriptions of all samples in the dataset. Specifically, we use `Gemini-1.5-Pro` to convert complex referring expressions into simpler ones. For example, the query "*Which food in the image is the richest in protein?*" is transformed into *"Eggs"*. For the **medium** and **hard** categories, we first run a preliminary evaluation using a baseline model (without any reasoning modules) and classify the samples into two groups: the top $50\%$ in terms of accuracy are labeled as **medium** difficulty, while the remaining $50\%$ are labeled as **hard** difficulty. This approach allows us to systematically analyze the impact of reasoning strategies across varying levels of query complexity.

## D MORE VISUAL GROUNDING RESULTS: REASONSEG

In Table 6, our method achieves the best performance on ReasonSeg benchmark. Following prior work, we denote "ft" as fine-tuning on the 236 training samples of ReasonSeg. The results indicate that fine-tuning significantly enhances model performance. Compared to RefCOCO, ReasonSeg features textual instructions that are often posed as questions, making them inherently more complex and requiring common-sense reasoning. Experimental results demonstrate that our model possesses strong reasoning capabilities and excels at handling complex reasoning scenarios.

## E HYPERPARAMETER SELECTION

The weighting coefficients $\lambda_1$ and $\lambda_2$ play a crucial role in balancing different components of our objective function. To determine the optimal values, we conducted a systematic grid search over the set $0.001, 0.01, 0.1, 1$ for each hyperparameter. We observed that setting $\lambda_1 = 0.1$ and $\lambda_2 = 0.1$ consistently yielded the best performance.

## F IMPLEMENTATION DETAILS

Training is conducted using the DeepSpeed engine for efficient large-scale optimization. We employ the AdamW Loshchilov & Hutter (2017) optimizer with a learning rate of 0.0003 and no weight

decay. The learning rate follows a WarmupDecayLR schedule with 100 warmup iterations. We use a per-device batch size of 2 with a gradient accumulation step of 10. For detailed hyperparameter settings, please refer to the *Appendices*.

## G   FALSE REJECTION EXAMPLES IN THE REJ MECHANISM

In Figure 5, two examples illustrate the concept of false rejection cases in the context of the `<REJ>` mechanism. On the left, the question asks about the presence of a fork in the image, and the system incorrectly outputs a `<REJ>` token despite the fork being clearly visible. This false rejection is likely due to occlusion, where part of the fork may be obscured by other objects, causing the model to incorrectly infer that the fork is absent or indistinct. Visual grounding mechanisms can sometimes struggle when key objects are partially hidden or not fully visible, leading to a misclassification.

On the right, the question inquires about the presence of an oven, with the system correctly rejecting the object. However, in this case, the failure occurs because the model incorrectly interprets the presence of an oven based on background objects or visual cues that resemble an oven but do not directly correspond to the target object. In this scenario, the model might rely too heavily on high-level features such as colors, textures, or spatial patterns that are similar to those typically associated with an oven, but not in this context. This could result in the model being more prone to hallucinations or misidentifying objects in cluttered or complex environments.

These examples demonstrate rare instances where the model falsely rejects an object that exists in the image, highlighting the importance of improving the precision of `<REJ>` outputs. To address these issues, refining the model's ability to handle occlusions and improve its attention to finer visual details could reduce false rejection rates, particularly in challenging cases where objects are partially visible or when background noise interferes with object recognition.

## H   MORE EXAMPLES

In Figure 6, we give more examples of different grounding choices: (a) The model selects the `<COMPLEX>` reasoning category for rationale generation before generating the final answer. (b) For simple queries, it selects the `<SIMPLE>` reasoning category to directly generate the final answer. (c) In visual grounding, the model selects the `<REJ>` category, returning a full background mask and a negative answer.

## I   LIMITATIONS AND DISCUSSION

A key design choice in our framework is the use of dataset-level `<SIMPLE>` and `<COMPLEX>` labels to support selective reasoning in textual grounding. While effective and scalable, this approximation is inherently coarse, as individual datasets may contain a mixture of simple and complex queries. This raises a natural concern about potential mismatches between dataset-level labels and the true per-sample reasoning complexity. We had already considered this issue during dataset construction, and our analysis indicates that this limitation has only a minimal effect on model performance for the following reasons:

**Dataset characteristics mitigate most mismatches.**   In practice, the proportion of mislabeled samples is very small because the datasets differ substantially in their native supervision format. Referring/dialogue datasets inherently exhibit multi-step reasoning, whereas detection-style datasets (e.g., COCO) contain only single-word category labels, which we template into simple sentences for training. As a result, dataset-level complexity largely aligns with the intrinsic reasoning structure of the samples.

**Sample-level relabeling experiment confirms minimal impact.**   To directly evaluate the reviewer's concern, we conducted a fine-grained relabeling experiment using an LLM-based pre-filter followed by human verification. We retrained PostAlign using these sample-level SIMPLE/COMPLEX labels. As reported in Table 7, the performance differences on POPE were marginal (e.g., 86.6 to 86.2 on the Random split), indicating that dataset-level labeling provides a sufficiently accurate approximation for our selective reasoning mechanism in practice.

Table 7: Ablation on complexity labeling. Despite the coarseness of dataset-level labeling, mixed-dataset training and soft gating make its practical impact minimal. Sample-level relabeling further confirms that fine-grained labels yield only marginal gains.

| Method | Ran | Pop | Adv |
|---|---|---|---|
| LLaVA-1.5-7B-PostAlign + unmixed-dataset | 85.3 | 83.4 | 81.7 |
| **LLaVA-1.5-7B-PostAlign + mixed-dataset (ours)** | **86.6** | **84.2** | **82.3** |
| LLaVA-1.5-7B-PostAlign + relabeled-dataset (sample-level) | 86.2 | 84.8 | 82.8 |

**Mixed-dataset training further reduces bias.** Our training pipeline always mixes heterogeneous datasets within each batch. This naturally places simple and complex reasoning patterns together during optimization, preventing the model from overfitting to dataset-level priors. This mixed training strategy is part of our original design and not introduced for rebuttal. For completeness, we additionally trained an *unmixed-dataset* variant, which indeed produced worse results in Table 7.

**Soft gating ensures robustness to borderline cases.** The SIMPLE/COMPLEX classifier outputs soft probabilities rather than hard labels. This soft gating mechanism allows PostAlign to adapt to borderline or mislabeled instances, reducing sensitivity to imperfect dataset-level supervision.

These results demonstrate that although dataset-level labeling is coarse, its practical impact on reasoning control is limited under our mixed-dataset training and soft-gating design. Fine-grained complexity labeling remains a promising direction for future work, particularly for large-scale heterogeneous multimodal corpora.

## J  BROADER SOCIETAL IMPACTS

Our proposed framework, MMGrounded-PostAlign, has the potential to positively impact a range of real-world applications that rely on multimodal understanding, such as assistive technologies for visually impaired users, educational tools, and human-AI collaboration in visual analytics. By enhancing visual grounding and reducing hallucinations, our method contributes to building more reliable and trustworthy AI systems that are better aligned with human expectations and safety standards.

However, as with many advanced AI systems, potential risks should be acknowledged. Improved grounding capabilities may inadvertently be misused in applications such as surveillance, automated misinformation generation, or biased decision-making if deployed without appropriate safeguards. Moreover, biases present in the training data could still propagate through the model, particularly in visually ambiguous or culturally sensitive contexts. It is important that future work continues to explore robust alignment, fairness auditing, and transparent usage policies to ensure responsible deployment of such technologies.

## K  THE USE OF LARGE LANGUAGE MODELS

No large language models (LLMs), including but not limited to ChatGPT, GPT-4, Claude, and Llama, were utilized in any stage of the research and writing process of this paper. All content presented in this work—encompassing the formulation of research questions, design of methodology, analysis of experimental data, drafting of the main text, compilation of references, and preparation of appendices—was independently conceived, developed, and written by the authors.

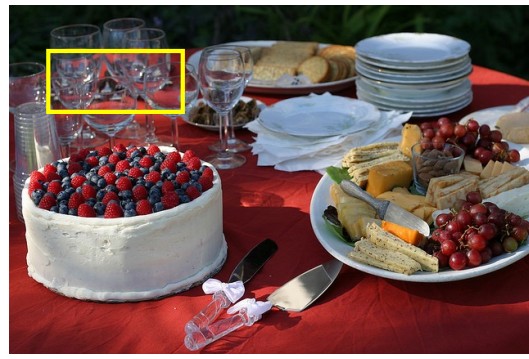 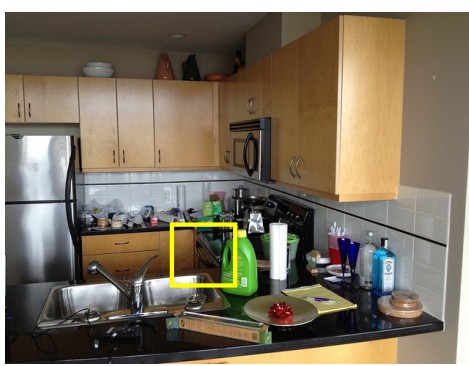

Is there a fork in the image?   Is there a oven in the image?

Figure 5: Two examples illustrate the concept of false rejection cases in the context of the <REJ> mechanism.

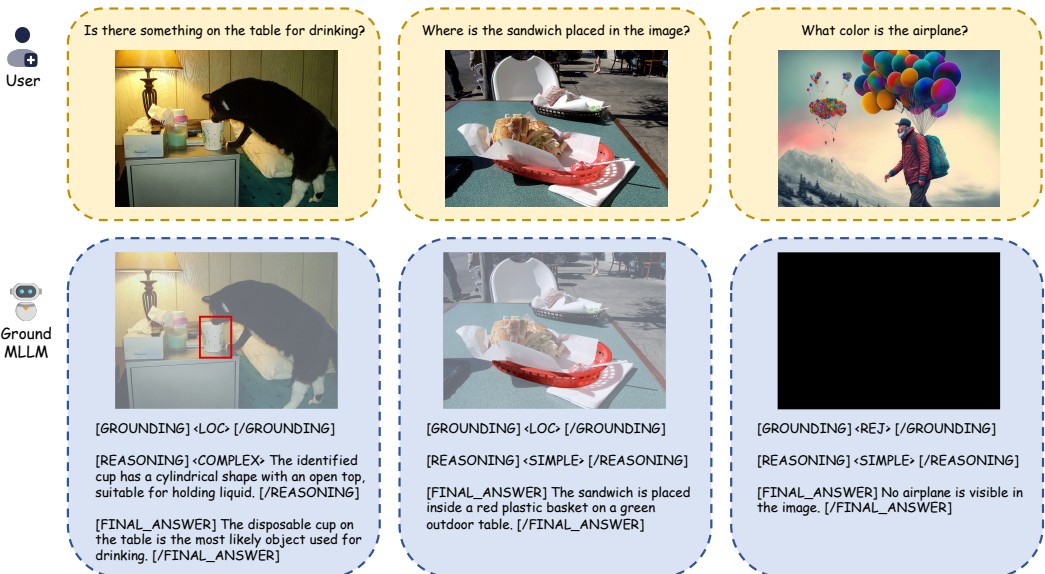

Figure 6: Examples of different grounding choices: (a) The model selects the <COMPLEX> reasoning category for rationale generation before generating the final answer. (b) For simple queries, it selects the <SIMPLE> reasoning category to directly generate the final answer. (c) In visual grounding, the model selects the <REJ> category, returning a full background mask and a negative answer.

