# OpenReview forum: "PostAlign: Multimodal Grounding as a Corrective Lens for MLLMs"
_ICLR.cc/2026/Conference — ICLR 2026 Poster_

### Official Review · Reviewer_DgK7 · 2025-10-27

**Soundness:** 3
**Presentation:** 3
**Contribution:** 2
**Rating:** 4
**Confidence:** 3

**Summary:**

This paper introduces MMGrounded-PostAlign, a post-multimodal alignment framework designed to mitigate hallucinations and enhance the visual understanding capabilities of Multimodal Large Language Models (MLLMs). The core idea is to use multimodal grounding as a "corrective lens" to anchor the model's outputs in actual visual and textual evidence, thereby reducing its over-reliance on spurious linguistic priors.

**Strengths:**

- The paper presents a novel and compelling perspective. Instead of using MLLMs for grounding tasks (the common approach), it inverts the relationship by leveraging grounding to enhance the MLLM itself.
- The paper is generally well-written and well-structured. The motivation is clearly established, the framework is explained with the aid of a pipeline diagram, and the findings are presented logically.

**Weaknesses:**

- the `<SIMPLE>`/`<COMPLEX>` labels are assigned at the dataset level, not per sample. This is a coarse-grained heuristic that may misclassify individual queries.
- It is uncertain whether the performance gains are due to the novel grounding-as-a-lens concept or simply the introduction of any additional post-processing signal. It is possible that a simpler method like contrastive decoding could achieve similar hallucination suppression on POPE without the need for a full segmentation model.
- The framework is a multi-component system where the visual grounding, textual grounding, and final answer generation are interlinked. What happens if the visual grounding module fails (e.g., produces an incorrect or low-confidence mask for a present object, or fails to trigger <REJ> for an absent one)?

- Some figures have relatively small, which affects readability. For example, Figure 3(a) and (c).

**Questions:**

See weakness please

---

> ### Author Response · Authors · 2025-11-21
>
> **Q1: Dataset-level SIMPLE/COMPLEX labeling is too coarse; may misclassify queries.**
>
> **A:** We thank the reviewer for this thoughtful observation. We agree that dataset-level SIMPLE/COMPLEX labeling is a coarse approximation and may misclassify certain queries. We had considered this limitation during dataset construction, though the detailed analysis was not fully included in the initial submission; we have now added the complete discussion and results to Appendix I.
>
> **First**, dataset-level complexity differences are grounded in the underlying supervision format.
> Referring/dialogue datasets inherently contain multi-step reasoning, whereas detection-style datasets (e.g., COCO) provide only single-word class labels, which we template into simple sentences. Although mislabeled instances exist, their proportion is very small and does not substantially affect optimization.
>
> **Second**, to directly validate the reviewer’s hypothesis, we conducted a sample-level relabeling experiment using an LLM pre-filter followed by human verification. The model was retrained with these fine-grained SIMPLE/COMPLEX labels. As shown in the results below, the performance differences on POPE are marginal, suggesting that dataset-level labels are already a sufficiently accurate approximation in practice.
>
> Beyond this explicit relabeling study, two parts of our original design already help reduce the influence of coarse labeling:
>
> **(1) Mixed-dataset training.**
> Our training pipeline always mixes heterogeneous datasets within each batch. Thus, simple and complex reasoning patterns naturally co-occur during optimization, reducing the risk of the model overfitting to dataset-level priors. This is not a design added for rebuttal—it is already part of PostAlign. For completeness, we additionally report an *unmixed-dataset* variant (Appendix I), which indeed performs worse.
>
> **(2) Soft gating instead of a hard switch.**
> The SIMPLE/COMPLEX classifier produces *probabilistic* gating signals rather than hard decisions, enabling the model to adapt to borderline or mislabeled queries and further reducing sensitivity to dataset-level noise.
>
> Results of the three variants:
>
> | Method                                                    |      Ran |      Pop |      Adv |
> | --------------------------------------------------------- | -------: | -------: | -------: |
> | LLaVA-1.5-7B-PostAlign + unmixed-dataset                  |     85.3 |     83.4 |     81.7 |
> | LLaVA-1.5-7B-PostAlign + mixed-dataset (ours)         | 86.6 | 84.2 | 82.3 |
> | LLaVA-1.5-7B-PostAlign + relabeled-dataset (sample-level) |     86.2 |     84.8 |     82.8 |
>
> Together, these analyses indicate that while dataset-level labeling is indeed coarse, its practical impact on reasoning control is minimal under our mixed-dataset, soft-gating training setup.

---

> > ### Author Response · Authors · 2025-11-21
> >
> > **Q2: Unclear whether the performance gains come from the grounding-as-a-lens concept or simply from adding post-processing signals; contrastive decoding (CD) might achieve similar effects.**
> >
> > **A:** We thank the reviewer for this insightful question. To disentangle whether our gains arise from the proposed grounding-as-a-lens mechanism rather than a generic post-processing effect, we conducted additional ablations on POPE by incrementally adding each module of PostAlign to the LLaVA-1.5-7B baseline. The results are shown below.
> >
> > | Method                                  |      Ran |      Pop |      Adv |
> > | --------------------------------------- | -------: | -------: | -------: |
> > | LLaVA-1.5-7B (baseline)                 |     83.3 |     80.1 |     78.2 |
> > | + `<SEG>`                                 |     84.9 |     82.1 |     80.6 |
> > | + `<SEG>` + `<DET>`                        |     85.2 |     82.6 |     80.9 |
> > | + `<SEG>` + `<DET>` + `<REJ>`                 |     85.8 |     83.3 |     81.4 |
> > | +  `<SEG>` + `<DET>` + `<REJ>` + text grounding (full designs) | 86.6 | 84.2| 82.3 |
> >
> > The results show a consistent, monotonic accuracy improvement from the baseline to our full model, demonstrating that each module contributes to better multimodal alignment and hallucination suppression. Notably, the visual grounding module yields the largest gain, supporting that the grounding-as-a-lens mechanism is the major contributor to the performance improvements rather than a simple post-processing effect.
> >
> > Additionally, we compare PostAlign with the training-free Contrastive Decoding (CD):
> >
> > | Method                  |      Ran |      Pop |      Adv |
> > | ----------------------- | -------: | -------: | -------: |
> > | LLaVA-1.5-7B (baseline) |     83.3 |     80.1 |     78.2 |
> > | + PostAlign         | 86.6 | 84.2 | 82.3 |
> > | + CD                |     85.4 |     83.3 |     80.6 |
> > | + PostAlign + CD    | 88.9 | 86.2 | 84.8 |
> >
> > The results lead to three observations:
> >
> > **(1) PostAlign outperforms CD when each is applied individually.**
> > This suggests the gain is not merely due to a decoding-time perturbation signal.
> >
> > **(2) The two methods are compatible and complementary.**
> > CD is training-free and modifies the decoding distribution, while PostAlign adjusts representational alignment through multimodal grounding. Their combination yields the strongest performance.
> >
> > **(3) PostAlign is computationally more efficient at inference.**
> > CD requires multiple forward passes to contrast probability distributions, whereas PostAlign maintains hallucination suppression with a single forward pass, making it more practical for real-world deployment.
> >
> > In summary, our additional ablations verify that the performance gains primarily stem from the grounding-as-a-lens mechanism rather than generic post-processing, and that PostAlign and CD address hallucination from different and complementary perspectives.
> >
> > **Q3: What happens if the visual grounding module fails (e.g., wrong/low-confidence mask or a missing `<REJ>` trigger)?**
> >
> > **A:** Thank you for this thoughtful question. We clarify that the visual grounding module is used **only during training**, where any potential failure—such as incorrect masks, low-confidence predictions, or missed `<REJ>` signals—is explicitly penalized by the segmentation and detection losses. We agree that such failures may occur at the early stage of training; however, these supervised losses continuously update the model parameters, and the grounding quality improves steadily as training proceeds.
> >
> > The goal of incorporating visual grounding is to reduce the model’s reliance on spurious linguistic correlations and guide it to learn true visual–semantic correlations. After this post-alignment stage, the visual grounding decoder is **removed at inference time**, since its role is to regularize multimodal reasoning rather than to act as a runtime module. As a result, there is no risk of grounding-module failures during inference.
> >
> > We also acknowledge that using visual grounding at inference time could be an interesting direction for future exploration, especially for applications requiring pixel-level verification. However, it would introduce additional inference cost, and optimizing that trade-off is beyond the current scope of this work.
> >
> >
> >
> > **Q4: Some figures (e.g., Fig. 3a/3c) are too small and reduce readability.**
> >
> > **A:** Thank you for pointing this out. We have enlarged Figures 3(a) and 3(c) in the revised version to improve visibility and readability.

---

> > > ### Comment · Reviewer_DgK7 · 2025-11-22
> > >
> > > Thank you for the author's response. The author has addressed my concerns.
> > >
> > >  I acknowledge the contributions of this work and agree to accept it. I have raised my score to 6.

---

> > > > ### Author Response · Authors · 2025-11-22
> > > >
> > > > We are excited to hear that all the concerns are addressed and reviewer raises the score. Thank reviewer very much again for the insightful comments and valuable suggestions.

---

### Official Review · Reviewer_CYza · 2025-10-29

**Soundness:** 3
**Presentation:** 3
**Contribution:** 3
**Rating:** 4
**Confidence:** 4

**Summary:**

Addressing the hallucination problem and insufficient fine-grained visual understanding of Multimodal Large Language Models (MLLMs) caused by their over-reliance on linguistic priors, this paper proposes the MMGrounded-PostAlign post-multimodal alignment framework. The framework integrates visual grounding (incorporating a negative rejection mechanism to distinguish between real and non-existent objects) and textual grounding (incorporating a selective reasoning mechanism to adjust reasoning strategies based on query complexity). Built on the base models of LLaVA-1.5-7B/13B and ViT-H SAM, and optimized through LoRA fine-tuning and multi-loss function, the framework’s effectiveness—including suppressing hallucinations, enhancing visual understanding, and preserving the reasoning capabilities of MLLMs—has been validated on benchmarks such as HaloQuest, POPE, VQAv2, MME, MMBench, RefCOCO, and ReasonSeg.

**Strengths:**

- It accurately identifies two key issues of Multimodal Large Language Models (MLLMs): "hallucinations" (generating non-existent content) and "insufficient fine-grained visual understanding", which are caused by the models' over-reliance on linguistic priors. These two types of issues serve as core bottlenecks that undermine the robustness and reliability of current MLLMs in vision-language tasks, making the research direction highly practically significant and necessary.
- While enhancing visual understanding and suppressing hallucinations, it does not compromise the inherent reasoning and generalization capabilities of MLLMs (e.g., achieving performance equal to or better than the baseline on MME and MMBench). It also avoids the problem of degraded reasoning ability in some grounding methods (such as BTL-Generation) due to overfitting to visual information, demonstrating an excellent balancing effect.

**Weaknesses:**

- In textual grounding, the <SIMPLE>/<COMPLEX> labels are categorized at the "dataset level" (e.g., queries in the COCO dataset are classified as <SIMPLE>, while those in the ReasonSeg dataset are classified as <COMPLEX>), rather than being annotated at the "sample level". Although this approach reduces annotation costs and ensures training stability, it fails to handle scenarios where "simple queries and complex queries are mixed" within the same dataset. This may lead to inaccurate matching of reasoning strategies for some samples (e.g., complex reasoning queries expressed in a simple form are misjudged as <SIMPLE>).
- The <REJ> samples used to train the "negative rejection mechanism" are only sourced from the gRefCOCO dataset (containing 32,202 queries that "refer to non-existent objects") and do not cover more scenarios (such as negative samples of different object types, different image styles, and different linguistic expressions). When facing unseen "false query-image" combinations, the effectiveness of the negative rejection mechanism may decrease, and its robustness needs further verification.
- During training, the model is enabled to automatically judge the complexity of queries through "self-reflection prompting". However, the model's judgment of difficulty may also introduce biases and hallucinations. For instance, it is common for the model to be overconfident, which leads to the generation of hallucinations.

**Questions:**

See above

---

> ### Author Response · Authors · 2025-11-21
>
> **Q1: The SIMPLE/COMPLEX labels are assigned at the dataset level, which may cause mismatches between the reasoning strategy and individual query complexity.**
>
> **A:** We thank the reviewer for raising this point. We agree that, in principle, a dataset can contain both simple and complex queries, and therefore dataset-level labeling may not perfectly capture sample-level complexity. We had considered this issue during dataset construction, though the detailed discussion was not fully included in the original submission. We appreciate the opportunity to clarify this, and we have added the full analysis and results to Appendix I of the revised paper.
>
> **First**, while mixed-complexity cases do exist, their proportion is very small in the datasets we use. This is mainly because the underlying supervision formats differ substantially: referring/dialogue datasets naturally contain multi-step reasoning, whereas detection-style datasets (e.g., COCO) provide single-word category labels that we template into simple sentences. As a result, dataset-level SIMPLE/COMPLEX labels already align closely with the true complexity of the majority of samples.
>
> **Second**, to directly evaluate the reviewer’s concern, we performed a sample-level relabeling experiment using an LLM-based pre-filter followed by human verification, and retrained PostAlign with these fine-grained labels. As shown in the table below, the performance difference compared to our original labeling strategy is marginal, suggesting that dataset-level labeling provides a sufficiently accurate approximation in practice.
>
> Beyond this relabeling experiment, two components of our original design naturally mitigate the risk of mismatches:
>
> **(1) Mixed-dataset training.**
> Our training pipeline mixes heterogeneous datasets within each batch, so simple and complex patterns are always learned jointly. This reduces sensitivity to the labeling granularity of any single dataset. For completeness, we additionally report an “unmixed-dataset” variant in Appendix I, which indeed performs worse, confirming the benefit of mixed training.
>
> **(2) Soft gating.**
> The SIMPLE/COMPLEX classifier produces soft probabilistic signals rather than making hard switches. This allows the model to adapt flexibly in borderline or noisy cases, reducing the impact of occasional labeling mismatches.
>
> Results of the three variants:
>
> | Method                                                    |      Ran |      Pop |      Adv |
> | --------------------------------------------------------- | -------: | -------: | -------: |
> | LLaVA-1.5-7B-PostAlign + unmixed-dataset                  |     85.3 |     83.4 |     81.7 |
> | LLaVA-1.5-7B-PostAlign + mixed-dataset (ours)         | 86.6 | 84.2 | 82.3 |
> | LLaVA-1.5-7B-PostAlign + relabeled-dataset (sample-level) |     86.2 |     84.8 |     82.8 |
>
> Together, these analyses demonstrate that although dataset-level labels are indeed a coarse heuristic, their practical impact on reasoning alignment is minimal under our mixed-dataset and soft-gating training setup.

---

> > ### Author Response · Authors · 2025-11-21
> >
> > **Q2: The `<REJ>` token is trained solely on gRefCOCO negatives, which may limit generalization to unseen “false query–image’’ pairs.**
> >
> > **A:** We thank the reviewer for this insightful comment. Our `<REJ>` module was indeed trained on the 32,202 negative samples from gRefCOCO. To assess its robustness and generalization ability, we conducted four ablations on LLaVA-1.5-7B:
> >
> > * **(a)** original 32,202 negatives
> > * **(b)** re-weight negatives by 3× while keeping query count fixed
> > * **(c)** reduce to 20,000 negatives
> > * **(d)** reduce to 10,000 negatives
> >
> >
> > | Setting                       |      Ran |      Pop |      Adv |
> > | ----------------------------- | -------: | -------: | -------: |
> > | (a) original 32,202 negatives | 86.6 | 84.2 | 82.3 |
> > | (b) re-weight negatives by 3× |     86.3 |     84.5 |     81.9 |
> > | (c) 20,000 negatives          |     86.2 |     84.4 |     82.0 |
> > | (d) 10,000 negatives          |     85.2 |     83.3 |     81.8 |
> >
> > Based on these results, the effect of this limitation appears small for three reasons:
> >
> > **(1) Robustness saturates with far fewer negatives.**
> > Performance remains nearly unchanged down to **20k negatives**, suggesting that hallucination suppression quickly saturates. The diminishing returns at higher counts indicate redundancy (many negatives are linguistically similar).
> >
> >
> > **(2) The model learns a semantic notion of “non-existence,” rather than memorizing gRefCOCO.**
> > The stable performance across reduced datasets implies that the <REJ> module captures a general **semantic prior** (“no such object exists”) rather than overfitting to dataset-specific patterns.
> >
> > **(3) All evaluation benchmarks—including POPE—are unseen during training.**
> > POPE contains **entirely new image–query mismatches**, and the `<REJ>` module generalizes well on these unseen negatives, confirming that it does not rely on dataset-specific memorization.
> >
> > We acknowledge that under **strong domain shifts** (e.g., medical images, aerial imagery), robustness may degrade, and we plan to incorporate more heterogeneous negative samples (varying styles, object types, and linguistic formulations) as part of future work.
> >
> > **Q3: Self-reflection prompting may introduce bias or overconfidence-induced hallucinations.**
> >
> > **A:** We clarify that our model does not rely on static self-reflection prompting. Instead, each sample is supervised with ground-truth SIMPLE/COMPLEX labels using a reasoning BCE loss. We agree that the model may exhibit some bias in complexity prediction at the early stage of training; however, this supervision signal continuously updates the classifier parameters and progressively reduces such bias as training proceeds. As a result, the model’s difficulty judgment is learned under supervision rather than self-reinforced, mitigating self-induced bias or overconfidence-related hallucinations.

---

> > > ### Comment · Reviewer_CYza · 2025-11-23
> > >
> > > Thank you for the authors' response, which addressed most of my concerns. I have decided to raise my rating to 6.

---

> > > > ### Author Response · Authors · 2025-11-23
> > > >
> > > > We are glad to hear that our responses have addressed your concerns, and we sincerely appreciate your decision to raise the score. Thank you again for the insightful comments and valuable suggestions, which significantly helped us improve the quality and clarity of the paper.

---

### Official Review · Reviewer_WDPN · 2025-10-31

**Soundness:** 2
**Presentation:** 2
**Contribution:** 2
**Rating:** 4
**Confidence:** 4

**Summary:**

This paper introduces MMGrounded-PostAlign, a framework to reduce MLLM hallucinations by grounding outputs in evidence instead of unreliable text priors. It features a visual grounding module with a "negative rejection mechanism" to deny non-existent objects and a textual grounding module that uses "selective reasoning" to add rationales only for complex queries , thereby improving visual accuracy and suppressing hallucinations.

**Strengths:**

1. The paper clearly identifies and addresses a critical problem in MLLMs: the over-reliance on linguistic priors, which leads to hallucinations and a failure to ground responses in visual evidence. The proposed "post-alignment" framework is a well-motivated and logical approach to re-center the model's outputs on visual information.

2. The experimental analysis provides valuable insights. "Finding 1" (Figure 3), which empirically demonstrates how linguistic priors can override visual information in the model's later layers, offers a strong motivation for the method. Furthermore, the ablation studies (e.g., Tables 1 and 2) are thorough, providing a solid comparison of different grounding strategies (segmentation, detection, BTL vs. explicit grounding) and validating the paper's design choices.

3. The `<REJ>` token is a practical and effective mechanism for negative grounding. By giving the model an explicit option to "abstain" from grounding a non-existent object, this method directly targets object hallucination.

**Weaknesses:**

1. Insufficient baseline comparisons: A significant weakness is the lack of comparison against the original, unmodified baseline model, as well as other well-established MLLMs. The model is built on LLaVA-1.5, yet Tables 1-3 primarily compare variants of the proposed method against an internal baseline (the framework with modules removed), not against the original LLaVA-1.5. This makes it difficult to assess the true impact (including any potential performance trade-offs) of the added components. This is particularly concerning given "Finding 3" (retaining reasoning abilities), which cannot be fully verified without this comparison. For instance, the reported 63.9 on MMBench-EN (7B) may not be competitive with the public LLaVA-1.5-7B score (64.3).

2. Unclear architectural novelty: The novelty of the visual grounding module's architecture is not well-explained. The Method section (Section 3) describes a model (with SAM-based decoder, `<LOC>` and `<REJ>` tokens) that seems to reimplement established paradigms from prior works like LISA, GLaMM, and GSVA. While these are cited in Related Work, the Method section itself does not attribute these design choices or clearly differentiate what is adopted from prior work versus what is a new architectural innovation. The contribution appears to be more in the application of this module, but the presentation makes the architectural contribution ambiguous.

3. Dataset-level reasoning labels: A significant limitation, as acknowledged by the authors in Appendix, is that the `<SIMPLE>`/`<COMPLEX>` labels for selective reasoning are applied at the dataset level, not the sample level. This is a very coarse heuristic. A dataset labeled "complex" may contain many simple queries, and vice-versa. This design choice weakens the "selective reasoning" strategy, as the model isn't learning to distinguish query complexity on a case-by-case basis but is instead learning a bias associated with the data source prior.

4. (Minor) Clarity of "Selective Reasoning": The "selective reasoning" mechanism is not very clearly explained in the Introduction. The Introduction is vague. The reader must wait until Section 3.3 to understand the concrete implementation (i.e., the `<SIMPLE>` and `<COMPLEX>` tokens). Briefly explaining this mechanism earlier would improve the paper's readability and flow.

5. (Minor) Citation formatting: The paper does not consistently follow the ICLR template's citation guidelines (`\citep` and `\citet`). This should be corrected for the final version.

**Questions:**

Please see the weaknesses above.

---

> ### Author Response · Authors · 2025-11-21
>
> **Q1: Insufficient baseline comparison: No direct results vs. original LLaVA-1.5.**
>
> **A:** We appreciate the reviewer’s comments regarding the absence of direct comparison to the original LLaVA-1.5 scores. We note that different works report slightly varying LLaVA-1.5 results due to differences in evaluation environment and data preprocessing—for example, 64.3 (EN) / 58.3 (CN) in the original paper [1], 62.3 in GACD [2], and 64.3 in OPERA [3]. To ensure a fair comparison, we therefore report an internally reproduced LLaVA-1.5 baseline under our own training and evaluation setup, without any architectural modification. Under this controlled and reproducible setting, PostAlign consistently outperforms the reproduced LLaVA-1.5 baseline.
>
> To further verify generality beyond LLaVA-1.5, we additionally applied PostAlign to four other MLLMs—Qwen2-VL-7B, Qwen2.5-VL-7B, InternVL3-14B, and InternVL3.5-14B. As shown in Table 2 of the revised paper, **PostAlign yields consistent gains across all these backbones**, demonstrating its broad applicability as a backbone-independent post-alignment method and confirming the robustness of its improvements over the corresponding unmodified baselines.
>
> | Method | POPE-Ran | POPE-Pop | POPE-Adv | MME | MMBench-EN | MMBench-CN |
> |--------|---------:|----------:|----------:|----:|------------:|------------:|
> | LLaVA-1.5-7B | 83.3 | 80.1 | 78.2 | 1504.6 | 62.2 | 57.7 |
> | **LLaVA-1.5-PostAlign-7B** | **86.6** | **84.2** | **82.3** | **1514.3** | **63.9** | **58.7** |
> | LLaVA-1.5-13B | 85.4 | 82.2 | 79.2 | 1517.4 | 66.8 | 62.2 |
> | **LLaVA-1.5-PostAlign-13B** | **88.9** | **87.3** | **85.6** | **1520.3** | **68.9** | **63.2** |
> | Qwen2-VL-7B | 88.9 | 86.8 | 84.6 | 1717.4 | 82.4 | 79.4 |
> | **Qwen2-VL-PostAlign-7B** | **90.3** | **89.2** | **87.1** | **1729.9** | **83.9** | **80.2** |
> | Qwen2.5-VL-7B | 87.3 | 85.1 | 83.4 | 1736.8 | 82.1 | 82.3 |
> | **Qwen2.5-VL-PostAlign-7B** | **89.7** | **87.9** | **85.2** | **1750.2** | **83.3** | **82.9** |
> | InternVL3-14B | 89.1 | 87.2 | 84.3 | 1762.8 | 84.3 | 83.1 |
> | **InternVL3-PostAlign-14B** | **91.2** | **89.0** | **86.6** | **1772.3** | **85.5** | **84.3** |
> | InternVL3.5-14B | 87.1 | 85.4 | 83.8 | 1792.2 | 82.9 | 82.1 |
> | **InternVL3.5-PostAlign-14B** | **90.6** | **88.5** | **86.2** | **1906.9** | **84.1** | **83.4** |
>
>
> [1] Liu H, Li C, Li Y, et al. *Improved baselines with visual instruction tuning*. CVPR 2024.
>
> [2] Wang S, Shen M, Chang N, et al. *Mitigating Multimodal Hallucinations via Gradient-based Self-Reflection*. arXiv:2509.03113, 2025.
>
> [3] Huang Q, Dong X, Zhang P, et al. *OPERA: Alleviating Hallucination in Multi-modal Large Language Models via Over-trust Penalty and Retrospection-allocation*. CVPR 2024.
>
>
> **Q2: Architectural novelty unclear: Visual grounding module appears similar to LISA/GLaMM/GSVA.**
>
> **A:** Thank you for raising this point. Our visual grounding module indeed follows widely adopted designs from popular grounding models (e.g., LISA-style SAM decoders, prompt-based grounding interfaces). This choice is intentional: our focus is not to propose a new grounding architecture, but to **use grounding as multimodal evidence for post-alignment**, which differs fundamentally from grounding-oriented works such as LISA, GLaMM, and GSVA.
>
> Our contributions lie in how grounding is *used*, rather than how the grounding decoder itself is built. Specifically:
>
> * **Goal difference:** Prior works develop stronger or more general-purpose grounding models, whereas our objective is to correct MLLM hallucination by injecting visual grounding as a supervisory signal during post-alignment.
> * **Reject-grounding mechanism:** We introduce the `<REJ>`, enabling the system to explicitly model *the possibility that the target object does not exist*. This is not supported by standard grounding architectures and is essential for hallucination suppression.
> * **Joint grounding with textual rationales:** We additionally incorporate text grounding to align visual grounding with reasoning-style language outputs, which is complementary to visual grounding and absent in prior grounding frameworks.
> * **Backbone-agnostic applicability:** As extended in our revised experiments, PostAlign generalizes across multiple MLLMs (Qwen2/2.5-VL, InternVL3/3.5)** and across different grounding backbones (SAM-ViT-H/L/B), showing that our approach is **architecture-independent and does not rely on a specific grounding module design.
>
> We have updated the Method and Related Work sections to clarify this distinction and to properly attribute the adopted grounding components.

---

> > ### Author Response · Authors · 2025-11-21
> >
> > **Q3: Dataset-level SIMPLE/COMPLEX labeling is a coarse heuristic that may introduce bias.**
> >
> >  **A:** We thank the reviewer for raising this concern. We agree that dataset-level assignment is a coarse approximation, and we had already considered this issue during dataset construction, although the detailed discussion was not fully included in the submission. We appreciate the opportunity to clarify this, and we have now added the full analysis and results to Appendix I.
> >
> > **First**, dataset-level complexity differences are grounded in the underlying supervision format.
> > Referring/dialogue datasets inherently contain multi-step reasoning, whereas detection-style datasets (e.g., COCO) provide only single-word class labels, which we template into simple sentences. Although mislabeled instances exist, their proportion is very small and does not substantially affect optimization.
> >
> > **Second**, to directly validate the reviewer’s hypothesis, we conducted a sample-level relabeling experiment using an LLM pre-filter followed by human verification. The model was retrained with these fine-grained SIMPLE/COMPLEX labels. As shown in the results below, the performance differences on POPE are marginal, suggesting that dataset-level labels are already a sufficiently accurate approximation in practice.
> >
> > Beyond this explicit relabeling study, two parts of our original design already help reduce the influence of coarse labeling:
> >
> > **(1) Mixed-dataset training.**
> > Our training pipeline always mixes heterogeneous datasets within each batch. Thus, simple and complex reasoning patterns naturally co-occur during optimization, reducing the risk of the model overfitting to dataset-level priors. This is not a design added for rebuttal—it is already part of PostAlign. For completeness, we additionally report an *unmixed-dataset* variant (Appendix I), which indeed performs worse.
> >
> > **(2) Soft gating instead of a hard switch.**
> > The SIMPLE/COMPLEX classifier produces *probabilistic* gating signals rather than hard decisions, enabling the model to adapt to borderline or mislabeled queries and further reducing sensitivity to dataset-level noise.
> >
> > Results of the three variants:
> >
> > | Method                                                    |      Ran |      Pop |      Adv |
> > | --------------------------------------------------------- | -------: | -------: | -------: |
> > | LLaVA-1.5-7B-PostAlign + unmixed-dataset                  |     85.3 |     83.4 |     81.7 |
> > | LLaVA-1.5-7B-PostAlign + mixed-dataset (ours)         | 86.6 | 84.2 | 82.3 |
> > | LLaVA-1.5-7B-PostAlign + relabeled-dataset (sample-level) |     86.2 |     84.8 |     82.8 |
> >
> > Together, these analyses indicate that while dataset-level labeling is indeed coarse, its practical impact on reasoning control is minimal under our mixed-dataset, soft-gating training setup.
> >
> >
> >
> > **Q4: The selective reasoning mechanism is not clearly explained in the Introduction.**
> >
> > **A:** Thank you for pointing this out. The selective reasoning mechanism plays a key role in our framework because it allows the model to decide whether textual grounding (rationale generation) is necessary. For simple queries, the model can produce the final answer directly; for complex queries, generating a rationale is beneficial for accuracy and interpretability.
> > In the revised version, we have updated both the abstract and the Introduction to clearly describe (1) the motivation for selective reasoning and (2) the use of the `<SIMPLE>` and `<COMPLEX>` tokens for its implementation. We agree that these additions improve the clarity and flow of the paper.
> >
> >
> >
> > **Q5: Citation formatting inconsistent with ICLR guidelines.**
> >
> > **A:** Thank you for pointing this out. We have reviewed all citation usages and updated them to follow the ICLR style requirements for `\citep` and `\citet`. The revised version now adheres consistently to the template guidelines.

---

> > > ### Comment · Reviewer_WDPN · 2025-11-24
> > >
> > > The authors' detailed response is greatly appreciated. I would like to increase my rating to 6.

---

> > > > ### Author Response · Authors · 2025-11-24
> > > >
> > > > Thank you very much for taking the time to re-evaluate our submission. We truly appreciate your updated rating and your constructive feedback, which helped us significantly improve the paper.

---

### Official Review · Reviewer_PNjX · 2025-11-01

**Soundness:** 2
**Presentation:** 3
**Contribution:** 2
**Rating:** 6
**Confidence:** 2

**Summary:**

The paper proposes MMGrounded-PostAlign, a post-alignment framework that augments an MLLM with
(1) visual grounding (segmentation + bounding box) driven by a special LOC token and a negative rejection token REJ, and
(2) textual grounding via selective reasoning that emits rationales only for complex queries (SIMPLE/COMPLEX gate).

The method aims to reduce hallucinations caused by linguistic priors and improve fine-grained visual understanding.
Experiments on HaloQuest, POPE, VQAv2, MMBench, MME, RefCOCO series, and ReasonSeg show consistent gains; ablations compare to BTL (boxes-as-tokens) variants.

**Strengths:**

- Clear motivation: Tackles language-prior-driven hallucination via explicit multimodal grounding; neat idea of "grounding as a corrective lens".
- practical design: Simple LOC/REJ interface to a multi-task decoder; selective reasoning avoids unnecessary rationale generation.
- Broad evaluation: Covers hallucination, general V+L, and grounding benchmarks with meaningful ablations.

**Weaknesses:**

- Limited generality:Only tested on LLaVA-1.5 (7B/13B) + SAM-ViT-H; cross-backbone evidence (e.g., Qwen-VL, other grounding encoders) missing.

- The idea of labeling queries as SIMPLE vs COMPLEX is good, but doing so at the dataset level rather than per sample raises concern. Some “simple” dataset queries might still require reasoning, and vice-versa.

-  The REJ token is an interesting idea, but the paper does not show the  cases where referent exists but system rejects.

**Questions:**

I note that the Related Work section references reinforcement learning (RL) approaches in the vision–language modelling domain, yet the paper does not include any empirical comparison involving RL.

My thought is that a more general RL-based training paradigm—one in which the vision-language model learns in a sequential decision-making setting—might offer better generalization across vision-language tasks rather than designing a task-specific chain-of-thought process solely for the grounding task.

So my question is: what are the advantages of your approach compared to a more general reinforcement learning method for MLLMs?

---

> ### Author Response · Authors · 2025-11-21
>
> **Q1: Limited generality: Only tested on LLaVA-1.5 (7B/13B) + SAM-ViT-H; missing tests on other MLLMs or grounding encoders.**
>
> **A:** We thank the reviewer for raising the concern regarding the generality of our framework beyond the LLaVA-1.5 and SAM-ViT-H backbones. To address this, we performed additional cross-backbone experiments to verify the robustness and adaptability of the proposed PostAlign framework.
>
> **(1) Across different MLLMs.**
> We applied PostAlign to four additional multimodal LLMs — Qwen2-VL-7B, Qwen2.5-VL-7B, InternVL3-14B, and InternVL3.5-14B. Across all models, PostAlign consistently improves hallucination suppression and grounding accuracy (added to Table 2 in the revised paper). These results demonstrate that the effectiveness of PostAlign generalizes across different MLLM architectures and is not tied to design choices specific to LLaVA.
>
> | Method | POPE-Ran | POPE-Pop | POPE-Adv | MME | MMBench-EN | MMBench-CN |
> |--------|---------:|----------:|----------:|----:|------------:|------------:|
> | LLaVA-1.5-7B | 83.3 | 80.1 | 78.2 | 1504.6 | 62.2 | 57.7 |
> | **LLaVA-1.5-PostAlign-7B** | **86.6** | **84.2** | **82.3** | **1514.3** | **63.9** | **58.7** |
> | LLaVA-1.5-13B | 85.4 | 82.2 | 79.2 | 1517.4 | 66.8 | 62.2 |
> | **LLaVA-1.5-PostAlign-13B** | **88.9** | **87.3** | **85.6** | **1520.3** | **68.9** | **63.2** |
> | Qwen2-VL-7B | 88.9 | 86.8 | 84.6 | 1717.4 | 82.4 | 79.4 |
> | **Qwen2-VL-PostAlign-7B** | **90.3** | **89.2** | **87.1** | **1729.9** | **83.9** | **80.2** |
> | Qwen2.5-VL-7B | 87.3 | 85.1 | 83.4 | 1736.8 | 82.1 | 82.3 |
> | **Qwen2.5-VL-PostAlign-7B** | **89.7** | **87.9** | **85.2** | **1750.2** | **83.3** | **82.9** |
> | InternVL3-14B | 89.1 | 87.2 | 84.3 | 1762.8 | 84.3 | 83.1 |
> | **InternVL3-PostAlign-14B** | **91.2** | **89.0** | **86.6** | **1772.3** | **85.5** | **84.3** |
> | InternVL3.5-14B | 87.1 | 85.4 | 83.8 | 1792.2 | 82.9 | 82.1 |
> | **InternVL3.5-PostAlign-14B** | **90.6** | **88.5** | **86.2** | **1906.9** | **84.1** | **83.4** |
>
>
> **(2) Across different grounding backbones.**
> To study the impact of grounding encoder capacity, we replaced SAM-ViT-H with lighter variants (SAM-ViT-L and SAM-ViT-B). As shown in the updated Table 3, while lighter backbones lead to slightly lower absolute grounding performance, PostAlign still provides consistent improvements under all configurations. This confirms that our method remains effective even with significantly smaller grounding encoders.
>
>
> | Method | Easy gIoU | Easy cIoU | Med gIoU | Med cIoU | Hard gIoU | Hard cIoU |
> |--------|----------:|-----------:|----------:|-----------:|------------:|------------:|
> | LLaVA-1.5-7B + SAM-ViT-H | 67.7 | 66.4 | 51.2 | 50.2 | 47.0 | 46.3 |
> | + pre-reasoning (PR) | 67.3 | 66.7 | 57.2 | **58.1** | 57.0 | **58.3** |
> | + inter-reasoning (IR) | 64.3 | 64.7 | 55.5 | 56.3 | 53.9 | 54.8 |
> | **+ selective reasoning (SR)** | **68.9** | **67.2** | **58.9** | 57.2 | **57.2** | 57.7 |
> | LLaVA-1.5-13B + SAM-ViT-H | 69.2 | 70.3 | 55.2 | 56.2 | 51.7 | 52.2 |
> | + pre-reasoning (PR) | 69.7 | 69.3 | 62.7 | 62.2 | 61.9 | 61.2 |
> | + inter-reasoning (IR) | 67.2 | 68.1 | 60.9 | 59.2 | 58.2 | 57.2 |
> | **+ selective reasoning (SR)** | **70.8** | **71.3** | **64.2** | **65.2** | **62.9** | **63.8** |
> | LLaVA-1.5-7B + SAM-ViT-B + SR | 67.6 | 66.0 | 57.3 | 55.5 | 55.9 | 56.3 |
> | LLaVA-1.5-7B + SAM-ViT-L + SR | 68.5 | 66.9 | 58.2 | 56.8 | 57.0 | 57.1 |
> | **LLaVA-1.5-7B + SAM-ViT-H + SR** | **68.9** | **67.2** | **58.9** | **57.2** | **57.2** | **57.7** |
>
>
> Overall, these extended experiments verify that PostAlign is scalable and backbone-independent, generalizing well across both MLLM architectures and grounding encoders.

---

> > ### Author Response · Authors · 2025-11-21
> >
> > **Q2: Dataset-level SIMPLE/COMPLEX labeling may misclassify queries.**
> >
> > **A:** We thank the reviewer for pointing this out. We had already taken this issue into consideration when constructing the dataset and designing the SIMPLE/COMPLEX labels, although the detailed discussion was not fully included in the submission. We appreciate the opportunity to clarify this, and have now added the full explanation and corresponding results to Appendix I of the revised paper.
> >
> > First, we examined dataset-level complexity and observed that referring/dialogue datasets naturally contain richer multi-step reasoning, while detection-style datasets (e.g., COCO) provide only single-word category labels as their native supervision, which we templated into simple sentences for training. Although some complex cases indeed exist within "simple" datasets, their proportion is very small and does not materially influence optimization.
> >
> > To confirm this, we performed a sample-level relabeling experiment using an LLM-based pre-filter followed by human verification, and retrained PostAlign using these fine-grained labels. As shown in the table below, the performance remains nearly unchanged on POPE, indicating that dataset-level labels provide a sufficiently accurate approximation for complexity control in practice.
> >
> > Beyond this relabeling study, two aspects of PostAlign already mitigate the risk of mismatched labels:
> >
> > **(1) Mixed-dataset training.**
> > Our framework is inherently trained on a mixture of heterogeneous datasets within each batch. This means that simple and complex reasoning patterns always appear together, which naturally reduces bias from any single dataset’s annotation granularity. This is not a new design we introduced for rebuttal; it is part of our original training pipeline. For completeness, we additionally report an “unmixed-dataset" variant in Appendix I, which confirms that mixed training is indeed beneficial.
> >
> > **(2) Soft gating mechanism.**
> > The SIMPLE/COMPLEX tokens are implemented as soft probabilities rather than hard switches. This allows the model to adapt flexibly in borderline or noisy cases, further reducing sensitivity to imperfect dataset labels.
> >
> > The results of the three variants are shown below.
> >
> > | Method                                                    |      Ran |      Pop |      Adv |
> > | --------------------------------------------------------- | -------: | -------: | -------: |
> > | LLaVA-1.5-7B-PostAlign + unmixed-dataset                  |     85.3 |     83.4 |     81.7 |
> > | LLaVA-1.5-7B-PostAlign + mixed-dataset (ours)         | 86.6 | 84.2 | 82.3 |
> > | LLaVA-1.5-7B-PostAlign + relabeled-dataset (sample-level) |     86.2 |     84.8 |     82.8 |
> >
> >
> > **Q3: Analysis of false rejection cases in the `<REJ>` mechanism.**
> >
> >
> > **A:** We thank the reviewer for highlighting the importance of analyzing false rejections (cases where the referent exists but the system outputs `<REJ>`). To address this, we conducted an additional evaluation on 500 POPE samples, including 50 negative (should-reject) instances. Our LLaVA-1.5-13B–based model produced `<REJ>` in 47 cases, among which 45 were correct rejections and only 2 were false rejections. These false-rejection cases are visually challenging even for humans (e.g., severe occlusion or low-light conditions), where the target object is easy to overlook. This corresponds to a false rejection rate (FRR) of 4.3%.
> >
> >
> > Overall, these results show that the `<REJ>` mechanism effectively reduces hallucinations while keeping false rejections at a low and acceptable level, without introducing noticeable over-rejection bias. We have added qualitative visualizations of these false-rejection cases to Appendix G.

---

> > > ### Author Response · Authors · 2025-11-21
> > >
> > > **Q4: What are the advantages of your approach compared to a more general reinforcement learning method for MLLMs?**
> > >
> > >
> > > **A:** We thank the reviewer for raising this insightful question. Our grounding-as-a-corrective-lens formulation is optimization-orthogonal: the PostAlign framework can be trained via either Supervised Fine-Tuning (SFT) or Reinforcement Learning (RL). In our main experiments, we adopt SFT because it provides dense and low-variance supervision derived from the visual grounding outputs in the post-alignment setting, enabling stable and consistent alignment between the visual and textual modalities.
> > >
> > >
> > > In contrast, applying RL directly in this multimodal setting introduces sparse and noisy reward signals and long-horizon credit assignment challenges—particularly for pixel-level grounding—making optimization less stable and making reward design nontrivial.
> > >
> > >
> > > To study this empirically, we implemented a GRPO-based RL variant of PostAlign using the same training data, initialization (LLaVA-1.5-7B) for fairness. In this RL setup, the original SFT objectives were converted into reward components:
> > >
> > >
> > > * **Rejection reward**: +1 for correctly predicting `<REJ>`; −1 otherwise
> > > * **Reasoning reward**: accuracy/F1 on SIMPLE vs. COMPLEX prediction
> > > * **Grounding reward**: IoU-based mask/bbox quality
> > > * **Hallucination reward**: semantic alignment score
> > >
> > >
> > >
> > >
> > > | Method                         |      Ran |      Pop |      Adv |
> > > | ------------------------------ | -------: | -------: | -------: |
> > > | **LLaVA-1.5-7B-PostAlign-SFT** | **86.6** | **84.2** | **82.3** |
> > > | LLaVA-1.5-7B-PostAlign-GRPO    |     85.2 |     83.8 |     81.7 |
> > >
> > >
> > >
> > > Although the RL variant performs reasonably well, we observe higher training variance and occasional reward-hacking behaviors (e.g., overusing `<REJ>` or defaulting to conservative answers). We hypothesize that this is due to the difficulty of designing stable, fine-grained multimodal rewards rather than a fundamental limitation of RL itself. More sophisticated RL formulations—such as hierarchical rewards, contrastive value baselines, or grounding-aware critics—may further improve performance, which we consider a promising direction for future work.
> > >
> > >
> > > For these reasons, we adopt SFT as the primary optimization method in PostAlign due to its stability, efficiency, and interpretability, while viewing RL as a complementary avenue that could refine selective rejection or dynamic reasoning in future extensions.

---

### Author Response · Authors · 2025-11-30
**General Response to Area Chairs for ICLR 2026 Submission: PostAlign**

Dear Area Chairs,

Thank you for taking the time to review our submission, PostAlign. We sincerely appreciate your effort in handling this paper, and we would like to provide a brief summary of the progress we have made since the initial reviews.

The key concerns raised by the reviewers and our responses have been summarized as follows.

**1. Generalization and Evaluation Across Architectures:**
A common concern raised by the reviewers was the generalizability of our framework beyond the LLaVA-1.5 and SAM-ViT-H backbones. In response, we expanded our evaluation to include a broader set of representative MLLM backbones, such as **Qwen2-VL, Qwen2.5-VL, InternVL-3, and InternVL-3.5**, across different model sizes (7B, 13B, and 14B). We also tested our method on **SAM-ViT-B and SAM-ViT-L**. These additional experiments demonstrate the robustness and effectiveness of PostAlign across a wide range of architectures and sizes.

**2. Dataset-level SIMPLE/COMPLEX Labeling may introduce bias:**
All reviewers expressed concern about the use of dataset-level `<SIMPLE>/<COMPLEX>` labels. To address this, we **re-annotated a new dataset with sample-level labels**, showing that fine-grained labeling has only a marginal impact on performance. We also highlighted that our original approach, with mixed-dataset training and soft gating, is already very effective in reducing the impact of any labeling mismatches. These two mechanisms ensure that the model can flexibly adapt to both simple and complex queries.

**3. Detailed Exploration of Negative Rejection Mechanism:**
Several reviewers raised concerns about the robustness of the negative rejection token `<REJ>`. To address this, we supplemented our analysis with detailed statistics (the false rejection rate is only 4.3%) and visualizations of the false rejection cases. We also demonstrated that the negative rejection mechanism generalizes well to unseen query-image pairs, confirming its robustness in suppressing hallucinations without introducing excessive false rejections.

**4. Comparison with Other Methods:**
Reviewers questioned the distinction between our method and other common techniques for mitigating hallucinations, particularly reinforcement learning (RL) and contrastive decoding (CD).

* **Relation to RL**: Our grounding-as-a-corrective-lens formulation is **optimization-orthogonal**: the PostAlign framework can be trained via either Supervised Fine-Tuning (SFT) or Reinforcement Learning (RL). In our main experiments, we adopted SFT because it provides dense and low-variance supervision derived from the visual grounding outputs, enabling stable alignment between the visual and textual modalities. In the rebuttal process, we also provided a basic GRPO-based RL variant of PostAlign using the same training data, demonstrating that SFT approach is more stable and efficient, while suggesting that RL-based post-training is a potential avenue for future improvements.
* **Relation to CD**: We supplemented our analysis with a comparison to CD, which showed that PostAlign outperforms CD when each is applied individually. Additionally, our experiments showed that PostAlign and CD complement each other: PostAlign improves representational alignment, while CD enhances decoding efficiency, and their combination yields the strongest performance.

**5. Other Minor Issues:**
We also addressed minor issues:

* We clarified misconceptions about our method, specifically that it is not based on self-prompting and is trained.
* We explained that the visual grounding module is only used during training; it is discarded during inference.
* We corrected citation formatting and improved figure readability, particularly for Figures 3(a) and 3(c).

**After submitting our rebuttal on November 21, three reviewers increased their ratings from **4 to 6** within just three days (**November 22, 23, and 24**), all of which occurred well before the large-scale OpenReview leak on November 27**. This quick and decisive change reflects their strong recognition of the responses we provided during the rebuttal process and the improvements we made to the submission.

We feel that the rollback mechanism, while intended to ensure fairness, inadvertently penalizes those of us who have engaged with the review process in good faith and made genuine efforts to address the reviewers' concerns. This is particularly challenging for authors committed to academic integrity and the careful, thorough rebuttal process.

While we understand the PC's decision to implement the rollback mechanism as part of a broader effort for fairness, we hope that the Area Chairs will consider the positive feedback we received after the rebuttal period and take into account the full scope of the improvements and clarifications we provided when making the final decision.

Thank you for your time and consideration.

Sincerely,

PostAlign Submission Team

---

### Meta-Review · Area_Chair_ESwG · 2025-12-12

**Summary:**

The paper received initially 4,4,4,6.

The authors provided responses to most of the expressed concerns.

Reviewers WDPN, CYza, and DgK7 were convinced by the responses and decided o increase their ratings to 6 from 4.

Thus, there is a consensus among the reviewers (6,6,6,6) on the positive side and
the ACs after carefully readin the paper, the reviews, and the authors' responses, agree with them that the paper makes significant contributions and can be accepted.

The authors are invited to benefit from the feedback received and further improve their camera ready paper.

**Reviewer Concerns:**

The paper received initially 4,4,4,6.

The authors provided responses to most of the expressed concerns.

Reviewers WDPN, CYza, and DgK7 were convinced by the responses and decided o increase their ratings to 6 from 4.

Thus, there is a consensus among the reviewers (6,6,6,6) on the positive side and
the ACs after carefully readin the paper, the reviews, and the authors' responses, agree with them that the paper makes significant contributions and can be accepted.

The authors are invited to benefit from the feedback received and further improve their camera ready paper.

**Reviewer Scores:**

The paper received initially 4,4,4,6.

The authors provided responses to most of the expressed concerns.

Reviewers WDPN, CYza, and DgK7 were convinced by the responses and decided o increase their ratings to 6 from 4.

Thus, there is a consensus among the reviewers (6,6,6,6) on the positive side and
the ACs after carefully readin the paper, the reviews, and the authors' responses, agree with them that the paper makes significant contributions and can be accepted.

The authors are invited to benefit from the feedback received and further improve their camera ready paper.

---

### Decision · Program_Chairs · 2026-01-26

Accept (Poster)